# Indexed Minimum Empirical Divergence-Based Algorithms for Linear Bandits

**Jie Bian**                                                                                  *jiebian@u.nus.edu*
*Department of Electrical and Computer Engineering*
*National University of Singapore*

**Vincent Y. F. Tan**                                                                         *vtan@nus.edu.sg*
*Department of Mathematics*
*Department of Electrical and Computer Engineering*
*National University of Singapore*

**Reviewed on OpenReview:** *https://openreview.net/forum?id=wE9kpJSemv*

## Abstract

The Indexed Minimum Empirical Divergence (IMED) algorithm is a highly effective approach that offers a stronger theoretical guarantee of the asymptotic optimality compared to the Kullback–Leibler Upper Confidence Bound (KL-UCB) algorithm for the multi-armed bandit problem. Additionally, it has been observed to empirically outperform UCB-based algorithms and Thompson Sampling. Despite its effectiveness, the generalization of this algorithm to contextual bandits with linear payoffs has remained elusive. In this paper, we present novel linear versions of the IMED algorithm, which we call the family of LinIMED algorithms. We demonstrate that LinIMED provides a $\widetilde{O}(d\sqrt{T})$ upper regret bound where $d$ is the dimension of the context and $T$ is the time horizon. Furthermore, extensive empirical studies reveal that LinIMED and its variants outperform widely-used linear bandit algorithms such as LinUCB and Linear Thompson Sampling in some regimes.

## 1 Introduction

The multi-armed bandit (MAB) problem (Lattimore & Szepesvári (2020)) is a classical topic in decision theory and reinforcement learning. Among the various subfields of bandit problems, the stochastic linear bandit is the most popular area due to its wide applicability in large-scale, real-world applications such as personalized recommendation systems (Li et al. (2010)), online advertising, and clinical trials. In the stochastic linear bandit model, at each time step $t$, the learner has to choose one arm $A_t$ from the time-varying action set $\mathcal{A}_t$. Each arm $a \in \mathcal{A}_t$ has a corresponding context $x_{t,a} \in \mathbb{R}^d$, which is a $d$-dimensional vector. By pulling the arm $a \in \mathcal{A}_t$ at time step $t$, under the linear bandit setting, the learner will receive the reward $Y_{t,a}$, whose expected value satisfies $\mathbb{E}[Y_{t,a}|x_{t,a}] = \langle \theta^*, x_{t,a} \rangle$, where $\theta^* \in \mathbb{R}^d$ is an unknown parameter. The goal of the learner is to maximize his cumulative reward over a time horizon $T$, which also means minimizing the cumulative regret, defined as $R_T := \mathbb{E}\left[\sum_{t=1}^{T} \max_{a \in \mathcal{A}_t} \langle \theta^*, x_{t,a} \rangle - Y_{t,A_t}\right]$. The learner needs to balance the trade-off between the exploration of different arms (to learn their expected rewards) and the exploitation of the arm with the highest expected reward based on the available data.

### 1.1 Motivation and Related Work

The $K$-armed bandit setting is a special case of the linear bandit. There exist several good algorithms such as UCB1 (Auer et al. (2002)), Thompson Sampling (Agrawal & Goyal (2012)), and the Indexed Minimum Empirical Divergence (IMED) algorithm (Honda & Takemura (2015)) for this setting. There are three main families of asymptotically optimal multi-armed bandit algorithms based on different principles (Baudry et al.

| | Problem independent regret bound | Regret bound independent of $K$? | Principle that the algorithm is based on |
|---|---|---|---|
| OFUL (Abbasi-Yadkori et al. (2011)) | $O(d\sqrt{T}\log(T))$ | ✓ | Optimism |
| LinUCB (Li et al. (2010)) | Hard to analyze | Unknown | Optimism |
| LinTS (Agrawal & Goyal (2013)) | $O(d^{\frac{3}{2}}\sqrt{T}) \wedge O(d\sqrt{T\log(K)})$ | ✓ | Posterior sampling |
| SupLinUCB (Chu et al. (2011)) | $O(\sqrt{dT\log^3(KT)})$ | ✗ | Optimism |
| LinUCB with OFUL's confidence bound | $O(d\sqrt{T}\log(T))$ | ✓ | Optimism |
| Asymptotically Optimal IDS (Kirschner et al. (2021)) | $O(d\sqrt{T}\log(T))$ | ✓ | Information directed sampling |
| LinIMED-3 (this paper) | $O(d\sqrt{T}\log(T))$ | ✓ | Min. emp. divergence |
| SupLinIMED (this paper) | $O(\sqrt{dT\log^3(KT)})$ | ✗ | Min. emp. divergence |

Table 1: Comparison of algorithms for linear bandits with varying arm sets

(2023)). However, among these algorithms, only IMED lacks an extension for contextual bandits with linear payoff. In the context of the varying arm setting of the linear bandit problem, the LinUCB algorithm in Li et al. (2010) is frequently employed in practice. It has a theoretical guarantee on the regret in the order of $O(d\sqrt{T}\log(T))$ using the confidence width as in OFUL (Abbasi-Yadkori et al. (2011)). Although the SupLinUCB algorithm introduced by Chu et al. (2011) uses phases to decompose the reward dependence of each time step and achieves an $\widetilde{O}(\sqrt{dT})$ (the $\widetilde{O}(\cdot)$ notation omits logarithmic factors in $T$) regret upper bound, its empirical performance falls short of both the algorithm in Li et al. (2010) and the Linear Thompson Sampling algorithm (Agrawal & Goyal (2013)) as mentioned in Lattimore & Szepesvári (2020, Chapter 22).

On the other hand, the Optimism in the Face of Uncertainty Linear (OFUL) bandit algorithm in Abbasi-Yadkori et al. (2011) achieves a regret upper bound of $\widetilde{O}(d\sqrt{T})$ through an improved analysis of the confidence bound using a martingale technique. However, it involves a bilinear optimization problem over the action set and the confidence ellipsoid when choosing the arm at each time. This is computationally expensive, unless the confidence ellipsoid is a convex hull of a finite set.

For randomized algorithms designed for the linear bandit problem, Agrawal & Goyal (2013) proposed the LinTS algorithm, which is in the spirit of Thompson Sampling (Thompson (1933)) and the confidence ellipsoid similar to that of LinUCB-like algorithms. This algorithm performs efficiently and achieves a regret upper bound of $O(d^{\frac{3}{2}}\sqrt{T} \wedge d\sqrt{T\log K})$, where $K$ is the number of arms at each time step such that $|\mathcal{A}_t| = K$ for all $t$. Compared to LinUCB with OFUL's confidence width, it has an extra $O(\sqrt{d} \wedge \sqrt{\log K})$ term for the minimax regret upper bound.

Recently, MED-like (minimum empirical divergence) algorithms have come to the fore since these randomized algorithms have the property that the probability of selecting each arm is in closed form, which benefits downstream work such as offline evaluation with the inverse propensity score. Both MED in the sub-Gaussian environment and its deterministic version IMED have demonstrated superior performances over Thompson Sampling (Bian & Jun (2021), Honda & Takemura (2015)). Baudry et al. (2023) also shows MED has a close relation to Thompson Sampling. In particular, it is argued that MED and TS can be interpreted as two variants of the same exploration strategy. Bian & Jun (2021) also shows that probability of selecting each arm of MED in the sub-Gaussian case can be viewed as a closed-form approximation of the same probability as in Thompson Sampling. We take inspiration from the extension of Thompson Sampling to linear bandits and thus are motivated to extend MED-like algorithms to the linear bandit setting and prove regret bounds that are competitive vis-à-vis the state-of-the-art bounds.

Thus, this paper aims to answer the question of whether it is possible to devise an extension of the IMED algorithm for the linear bandit problem the varying arm set setting (for both infinite and finite arm sets) with a regret upper bound of $O(d\sqrt{T}\log T)$ which matches LinUCB with OFUL's confidence bound while being as efficient as LinUCB. The proposed family of algorithms, called LinIMED as well as SupLinIMED, can be viewed as generalizations of the IMED algorithm (Honda & Takemura (2015)) to the linear bandit setting. We prove that LinIMED and its variants achieve a regret upper bound of $\widetilde{O}(d\sqrt{T})$ and they perform

efficiently, no worse than LinUCB. SupLinIMED has a regret bound of $\widetilde{O}(\sqrt{dT})$, but works only for instances with finite arm sets. In our empirical study, we found that the different variants of LinIMED perform better than LinUCB and LinTS for various synthetic and real-world instances under consideration.

Compared to OFUL, LinIMED works more efficiently. Compared to SupLinUCB, our LinIMED algorithm is significantly simpler, and compared to LinUCB with OFUL's confidence bound, our empirical performance is better. This is because in our algorithm, the exploitation term and exploration term are decoupling and this leads to a finer control while tuning the hyperparameters in the empirical study.

Compared to LinTS, our algorithm's (specifically LinIMED-3) regret bound is superior, by an order of $O(\sqrt{d} \wedge \sqrt{\log K})$. Since fixed arm setting is a special case of finite varying arm setting, our result is more general than other fixed-arm linear bandit algorithms like Spectral Eliminator (Valko et al. (2014)) and PEGOE (Lattimore & Szepesvári (2020, Chapter 22)). Finally, we observe that since the index used in LinIMED has a similar form to the index used in the Information Directed Sampling (IDS) procedure in Kirschner et al. (2021) (which is known to be asymptotically optimal but more difficult to compute), LinIMED performs significantly better on the "End of Optimism" example in Lattimore & Szepesvari (2017). We summarize the comparisons of LinIMED to other linear bandit algorithms in Table 1. We discussion comparisons to other linear bandit algorithms in Sections 3.2, 3.3, and Appendix B.

## 2 Problem Statement

**Notations:** For any $d$ dimensional vector $x \in \mathbb{R}^d$ and a $d \times d$ positive definite matrix $A$, we use $\|x\|_A$ to denote the Mahalanobis norm $\sqrt{x^\top A x}$. We use $a \wedge b$ (resp. $a \vee b$) to represent the minimum (resp. maximum) of two real numbers $a$ and $b$.

**The Stochastic Linear Bandit Model:** In the stochastic linear bandit model, the learner chooses an arm $A_t$ at each round $t$ from the arm set $\mathcal{A}_t = \{a_{t,1}, a_{t,2}, \ldots\} \subseteq \mathbb{R}$, where we assume the cardinality of each arm set $\mathcal{A}_t$ can be potentially infinite such that $|\mathcal{A}_t| = \infty$ for all $t \geq 1$. Each arm $a \in \mathcal{A}_t$ at time $t$ has a corresponding context (arm vector) $x_{t,a} \in \mathbb{R}^d$, which is known to the learner. After choosing arm $A_t$, the environment reveals the reward

$$Y_t = \langle \theta^*, X_t \rangle + \eta_t$$

to the learner where $X_t := x_{t,A_t}$ is the corresponding context of the arm $A_t$, $\theta^* \in \mathbb{R}^d$ is an unknown coefficient of the linear model, $\eta_t$ is an $R$-sub-Gaussian noise conditioned on $\{A_1, A_2, \ldots, A_t, Y_1, Y_2, \ldots, Y_{t-1}\}$ such that for any $\lambda \in \mathbb{R}$, almost surely,

$$\mathbb{E}\left[\exp(\lambda \eta_t) \mid A_1, A_2, \ldots, A_t, Y_1, Y_2, \ldots, Y_{t-1}\right] \leq \exp\left(\frac{\lambda^2 R^2}{2}\right).$$

Denote $a_t^* := \arg\max_{a \in \mathcal{A}_t} \langle \theta^*, x_{t,a} \rangle$ as the arm with the largest reward at time $t$. The goal is to minimize the expected cumulative regret over the horizon $T$. The (expected) cumulative regret is defined as

$$R_T = \mathbb{E}\left[\sum_{t=1}^T \langle \theta^*, x_{t,a_t^*} \rangle - \langle \theta^*, X_t \rangle\right].$$

**Assumption 1.** *For each time $t$, we assume that $\|X_t\| \leq L$, and $\|\theta^*\| \leq S$ for some fixed $L, S > 0$. We also assume that $\Delta_{t,b} := \max_{a \in \mathcal{A}_t} \langle \theta^*, x_{t,a} \rangle - \langle \theta^*, x_{t,b} \rangle \leq 1$ for each arm $b \in \mathcal{A}_t$ and time $t$.*

## 3 Description of LinIMED Algorithms

In the pseudocode of Algorithm 1, for each time step $t$, in Line 4, we use the improved confidence bound of $\theta^*$ as in Abbasi-Yadkori et al. (2011) to calculate the confidence bound $\beta_{t-1}(\gamma)$. After that, for each arm $a \in \mathcal{A}_t$, in Lines 6 and 7, the empirical gap between the highest empirical reward and the empirical reward of arm $a$ is estimated as

$$\hat{\Delta}_{t,a} = \begin{cases} \max_{j \in \mathcal{A}_t} \langle \hat{\theta}_{t-1}, x_{t,j} \rangle - \langle \hat{\theta}_{t-1}, x_{t,a} \rangle & \text{if LinIMED-1,2} \\ \max_{j \in \mathcal{A}_t} \mathrm{UCB}_t(j) - \mathrm{UCB}_t(a) & \text{if LinIMED-3} \end{cases}$$

---

**Algorithm 1** LinIMED-$x$ for $x \in \{1, 2, 3\}$

---

1: **Input:** LinIMED mode $x$, Dimension $d$, Regularization parameter $\lambda$, Bound $S$ on $\|\theta^*\|$, Sub-Gaussian parameter $R$, Concentration parameter $\gamma$ of $\theta^*$, Bound $L$ on $\|x_{t,a}\|$ for all $t \geq 1$ and $a \in \mathcal{A}_t$, Constant $C \geq 1$.

2: **Initialize:** $V_0 = \lambda I_{d \times d}$, $W_0 = 0_{d \times 1}$(all zeros vector with $d$ dimensions), $\hat{\theta}_0 = V_0^{-1}W_0$

3: **for** $t = 1, 2, \ldots T$ **do**

4:      Receive the arm set $\mathcal{A}_t$ and compute $\beta_{t-1}(\gamma) = \left( R\sqrt{d\log(\frac{1+(t-1)L^2/\lambda}{\gamma})} + \sqrt{\lambda}S \right)^2$.

5:      **for** $a \in \mathcal{A}_t$ **do**

6:          Compute: $\hat{\mu}_{t,a} = \langle \hat{\theta}_{t-1}, x_{t,a} \rangle$ and $\mathrm{UCB}_t(a) = \langle \hat{\theta}_{t-1}, x_{t,a} \rangle + \sqrt{\beta_{t-1}(\gamma)}\|x_{t,a}\|_{V_{t-1}^{-1}}$

7:          Compute: $\hat{\Delta}_{t,a} = (\max_{j \in \mathcal{A}_t} \hat{\mu}_{t,j} - \hat{\mu}_{t,a}) \cdot \mathbb{1}\{x = 1, 2\} + (\max_{j \in \mathcal{A}_t} \mathrm{UCB}_t(j) - \mathrm{UCB}_t(a)) \cdot \mathbb{1}\{x = 3\}$

8:          **if** $a = \arg\max_{j \in \mathcal{A}_t}(\hat{\mu}_{t,j} \cdot \mathbb{1}\{x = 1, 2\} + \mathrm{UCB}_t(j) \cdot \mathbb{1}\{x = 3\})$ **then**

9:          $I_{t,a} = -\log(\beta_{t-1}(\gamma)\|x_{t,a}\|_{V_{t-1}^{-1}}^2) \cdot \mathbb{1}\{x = 1\}$                                 (LinIMED-1)

10:             $+ \left[ \log T \wedge \left( -\log(\beta_{t-1}(\gamma)\|x_{t,a}\|_{V_{t-1}^{-1}}^2) \right) \right] \cdot \mathbb{1}\{x = 2\}$         (LinIMED-2)

11:             $+ \left[ \log \frac{C}{\max_{a \in \mathcal{A}_t} \hat{\Delta}_{t,a}^2} \wedge \left( -\log(\beta_{t-1}(\gamma)\|x_{t,a}\|_{V_{t-1}^{-1}}^2) \right) \right] \cdot \mathbb{1}\{x = 3\}$     (LinIMED-3)

12:          **else**

13:          $I_{t,a} = \frac{\hat{\Delta}_{t,a}^2}{\beta_{t-1}(\gamma)\|x_{t,a}\|_{V_{t-1}^{-1}}^2} - \log(\beta_{t-1}(\gamma)\|x_{t,a}\|_{V_{t-1}^{-1}}^2)$

14:          **end if**

15:      **end for**

16:      Pull the arm $A_t = \arg\min_{a \in \mathcal{A}_t} I_{t,a}$ (ties are broken arbitrarily) and receive its reward $Y_t$.

17:      **Update:**

18:      $V_t = V_{t-1} + X_t X_t^\top$, $W_t = W_{t-1} + Y_t X_t$, and $\hat{\theta}_t = V_t^{-1}W_t$.

19: **end for**

---

Then, in Lines 9 to 11, with the use of the confidence width of $\beta_{t-1}(\gamma)$, we can compute the index $I_{t,a}$ for the empirical best arm $a = \arg\max_{j \in \mathcal{A}_t} \hat{\mu}_{t,a}$ (for LinIMED-1,2) or the highest UCB arm $a = \arg\max_{j \in \mathcal{A}_t} \mathrm{UCB}_j(a)$ (for LinIMED-3). The different versions of LinIMED encourage different amounts of exploitation. For the other arms, in Line 13, the index is defined and computed as

$$I_{t,a} = \frac{\hat{\Delta}_{t,a}^2}{\beta_{t-1}(\gamma)\|x_{t,a}\|_{V_{t-1}^{-1}}^2} + \log \frac{1}{\beta_{t-1}(\gamma)\|x_{t,a}\|_{V_{t-1}^{-1}}^2}.$$

Then with all the indices of the arms calculated, in Line 16, we choose the arm $A_t$ with the minimum index such that $A_t = \arg\min_{a \in \mathcal{A}_t} I_{t,a}$ (where ties are broken arbitrarily) and the agent receives its reward. Finally, in Line 18, we use ridge regression to estimate the unknown $\theta^*$ as $\hat{\theta}_t$ and update the matrix $V_t$ and the vector $W_t$. After that, the algorithm iterates to the next time step until the time horizon $T$. From the pseudo-code, we observe that the only differences between the three algorithms are the way that the square gap, which plays the role of the empirical divergence, is estimated and the index of the empirically best arm. The latter point implies that we encourage the empirically best arm to be selected more often in LinIMED-2 and LinIMED-3 compared to LinIMED-1; in other words, we encourage more exploitation in LinIMED-2 and LinIMED-3. Similar to the core spirit of IMED algorithm Honda & Takemura (2015), the first term of our index $I_{t,a}$ for LinIMED-1 algorithm is $\hat{\Delta}_{t,a}^2/(\beta_{t-1}(\gamma)\|x_{t,a}\|_{V_{t-1}^{-1}}^2)$, this is the term controls the exploitation, while the second term $-\log(\beta_{t-1}(\gamma)\|x_{t,a}\|_{V_{t-1}^{-1}}^2)$ controls the exploration in our algorithm.

### 3.1 Description of the SupLinIMED Algorithm

Now we consider the case in which the arm set $\mathcal{A}_t$ at each time $t$ is finite but still time-varying. In particular, $\mathcal{A}_t = \{a_{t,1}, a_{t,2}, \ldots, a_{t,K}\} \subseteq \mathbb{R}$ are sets of constant size $K$ such that $|\mathcal{A}_t| = K < \infty$. In the pseudocode of Algorithm 2, we apply the SupLinUCB framework (Chu et al., 2011), leveraging Algorithm 3 (in Appendix A)

---

**Algorithm 2** SupLinIMED

---

1: **Input:** $T \in \mathbb{N}$, $S' = \lceil \log T \rceil$, $\Psi_t^s = \emptyset$ for all $s \in [S'], t \in [T]$
2: **for** $t = 1, 2, \ldots, T$ **do**
3:    $s \leftarrow 1$ and $\hat{\mathcal{A}}_1 \leftarrow [K]$
4:   **repeat**
5:      Use BaseLinUCB (stated in Algorithm 3 in Appendix A) with $\Psi_t^s$ to calculate the width $w_{t,a}^s$ and
       sample mean $\hat{Y}_{t,a}^s$ for all $a \in \hat{\mathcal{A}}_s$ .
6:      **if** $w_{t,a}^s \leq \frac{1}{\sqrt{T}}$ for all $a \in \hat{\mathcal{A}}_s$ **then**
7:        choose $A_t = \arg\min_{a \in \hat{\mathcal{A}}_s} I_{t,a}$ where $I_{t,a}$ is the same index function as in LinIMED algorithm:
8:        Calculate the index

$$I_{t,a} = \begin{cases} \log(2T) \wedge \left(-\log((w_{t,a}^s)^2)\right) & \textbf{If } a = \arg\max_{b \in \hat{\mathcal{A}}_s} \hat{Y}_{t,b}^s \\ (\frac{\hat{\Delta}_{t,a}^s}{w_{t,a}^s})^2 - \log((w_{t,a}^s)^2) & \textbf{otherwise} \end{cases} \quad \text{where} \quad \hat{\Delta}_{t,a}^s := \max_{b \in \hat{\mathcal{A}}_s} \hat{Y}_{t,b}^s - \hat{Y}_{t,a}^s .$$

9:        Keep the same index sets at all levels: $\Psi_{t+1}^{s'} \leftarrow \Psi_t^{s'}$ for all $s' \in [S]$ .         $\leftarrow$ **Case 1**
10:      **else if** $w_{t,a}^s \leq 2^{-s}$ for all $a \in \hat{\mathcal{A}}_s$ **then**
11:        $\hat{\mathcal{A}}_{s+1} \leftarrow \left\{ a \in \hat{\mathcal{A}}_s : \hat{Y}_{t,a}^s + w_{t,a}^s \geq \max_{a' \in \hat{\mathcal{A}}_s}(\hat{Y}_{t,a'}^s + w_{t,a'}^s) - 2^{1-s} \right\}$
12:        $s \leftarrow s + 1$                            $\leftarrow$ **Case 2**
13:      **else**
14:        Choose $A_t \in \hat{\mathcal{A}}_s$ such that $w_{t,A_t}^s > 2^{-s}$
15:        Update the index sets at all levels: $\Psi_{t+1}^{s'} \leftarrow \Psi_t^{s'} \cup \{t\}$ if $s = s'$ ; $\Psi_{t+1}^{s'} \leftarrow \Psi_t^{s'}$ if $s \neq s'$   $\leftarrow$ **Case 3**
16:      **end if**
17:   **until** an action $A_t$ is found
18: **end for**

---

as a subroutine within each phase. This ensures the independence of the choice of the arm from past observations of rewards, thereby yielding a concentration inequality in the estimated reward (see Lemma 1 in Chu et al. (2011)) that converges to within $\sqrt{d}$ proximity of the unknown expected reward in a finite arm setting. As a result, the regret yields an improvement of $\sqrt{d}$ ignoring the logarithmic factor. At each time step $t$ and phase $s$, in Line 5, we utilize the BaseLinUCB Algorithm as a subroutine to calculate the sample mean and confidence width since we also need these terms to calculate the IMED-style indices of each arm. In Lines 6–9 (Case 1), if the width of each arm is less than $\frac{1}{\sqrt{T}}$, we choose the arm with the smaller IMED-style index. In Lines 10–12 (Case 2), the framework is the same as in SupLinUCB (Chu et al. (2011)), if the width of each arm is smaller than $2^{-s}$ but there exist arms with widths larger than $\frac{1}{\sqrt{T}}$, then in Line 11 the "unpromising" arms will be eliminated until the width of each arm is smaller enough to satisfy the condition in Line 6. Otherwise, if there exist any arms with widths that are larger than $2^{-s}$, in Lines 14–15 (Case 3), we choose one such arm and record the context and reward of this arm to the next layer $\Psi_{t+1}^s$.

### 3.2 Relation to the IMED algorithm of Honda & Takemura (2015)

The IMED algorithm is a deterministic algorithm for the $K$-armed bandit problem. At each time step $t$, it chooses the arm $a$ with the minimum index, i.e.,

$$a = \arg\min_{i \in [K]} \left\{ T_i(t) D_{\inf}(\hat{F}_i(t), \hat{\mu}^*(t)) + \log T_i(t) \right\} , \tag{1}$$

where $T_i(t) = \sum_{s=1}^{t-1} \mathbb{1}\{A_t = a\}$ is the total arm pulls of the arm $i$ until time $t$ and $D_{\inf}(\hat{F}_i(t), \hat{\mu}^*(t))$ is some divergence measure between the empirical distribution of the sample mean for arm $i$ and the arm with the highest sample mean. More precisely, $D_{\inf}(F, \mu) := \inf_{G \in \mathcal{G}:\mathbb{E}(G) \leq \mu} D(F \| G)$ and $\mathcal{G}$ is the family of distributions supported on $(-\infty, 1]$. As shown in Honda & Takemura (2015), its asymptotic bound is even better than KL-UCB (Garivier & Cappé (2011)) algorithm and can be extended to semi-bounded support models such as $\mathcal{G}$. Also, this algorithm empirically outperforms the Thompson Sampling algorithm as shown

in Honda & Takemura (2015). However, an extension of IMED algorithm with minimax regret bound of $\widetilde{O}(d\sqrt{T})$ has not been derived. In our design of LinIMED algorithm, we replace the optimized KL-divergence measure in IMED in Eqn. (1) with the squared gap between the sample mean of the arm $i$ and the arm with the maximum sample mean. This choice simplifies our analysis and does not adversely affect the regret bound. On the other hand, we view the term $1/T_i(t)$ as the variance of the sample mean of arm $i$ at time $t$; then in this spirit, we use $\beta_{t-1}(\gamma)\|x_{t,a}\|_{V_{t-1}^{-1}}^2$ as the variance of the sample mean (which is $\langle\hat{\theta}_{t-1}, x_{t,a}\rangle$) of arm $a$ at time $t$. We choose $\hat{\Delta}_{t,a}^2/(\beta_{t-1}(\gamma)\|x_{t,a}\|_{V_{t-1}^{-1}}^2)$ instead of the KL-divergence approximation for the index since in the classical linear bandit setting, the noise is sub-Gaussian and it is known that the KL-divergence of two Gaussian random variables with the same variance $(\mathrm{KL}(\mathcal{N}(\mu_1, \sigma^2), \mathcal{N}(\mu_2, \sigma^2)) = \frac{(\mu_1-\mu_2)^2}{2\sigma^2})$ has a closed form expression similar to $\hat{\Delta}_{t,a}^2/(\beta_{t-1}(\gamma)\|x_{t,a}\|_{V_{t-1}^{-1}}^2)$ ignoring the constant $\frac{1}{2}$.

### 3.3 Relation to Information Directed Sampling (IDS) for Linear Bandits

Information Directed Sampling (IDS), introduced by Russo & Van Roy (2014), serves as a good principle for regret minimization in linear bandits to achieve the asymptotic optimality. The intuition behind IDS is to balance between the information gain on the best arm and the expected reward at each time step. This goal is realized by optimizing the distribution of selecting each arm $\pi \in \mathcal{D}(\mathcal{A})$ (where $\mathcal{A}$ is the fixed finite arm set) with the minimum information ratio such that:

$$\pi_t^{\mathrm{IDS}} := \arg\min_{\pi\in\mathcal{D}(\mathcal{A})} \frac{\hat{\Delta}_t^2(\pi)}{g_t(\pi)},$$

where $\hat{\Delta}_t$ is the empirical gap and $g_t$ is the so-called information gain (defined later). Kirschner & Krause (2018), Kirschner et al. (2020) and Kirschner et al. (2021) apply the IDS principle to the linear bandit setting, The first two works propose both randomized and deterministic versions of IDS for linear bandit. They showed a near-optimal minimax regret bound of the order of $\widetilde{O}(d\sqrt{T})$. Kirschner et al. (2021) designed an asymptotically optimal linear bandit algorithm retaining its near-optimal minimax regret properties. Comparing these algorithms with our LinIMED algorithms, we observe that the first term of the index of non-greedy actions in our algorithms is $\hat{\Delta}_{t,a}^2/(\beta_{t-1}(\gamma)\|x_{t,a}\|_{V_{t-1}^{-1}}^2)$, which is similar to the choice of information ratio in IDS with the estimated gap $\Delta_t(a) := \hat{\Delta}_{t,a}$ as defined in Algorithm 1 and the information ratio $g_t(a) := \beta_{t-1}(\gamma)\|x_{t,a}\|_{V_{t-1}^{-1}}^2$. As mentioned in Kirschner & Krause (2018), when the noise is 1-subgaussian and $\|x_{t,a}\|_{V_{t-1}^{-1}}^2 \ll 1$, the information gain in deterministic IDS algorithm is approximately $\|x_{t,a}\|_{V_{t-1}^{-1}}^2$, which is similar to our choice $\beta_{t-1}(\gamma)\|x_{t,a}\|_{V_{t-1}^{-1}}^2$. However, our LinIMED algorithms are different from the deterministic IDS algorithm in Kirschner & Krause (2018) since the estimated gap defined in our algorithm $\hat{\Delta}_{t,a}$ is different from that in deterministic IDS. Furthermore, as discussed in Kirschner et al. (2020), when the noise is 1-subgaussian and $\|x_{t,a}\|_{V_{t-1}^{-1}}^2 \ll 1$, the action chosen by UCB minimizes the deterministic information ratio. However, this is not the case for our algorithm since we have the second term $-\log(\beta_{t-1}(\gamma)\|x_{t,a}\|_{V_{t-1}^{-1}}^2)$ in LinIMED-1 which balances information and optimism. Compared to IDS in Kirschner et al. (2021), their algorithm is a randomized version of the deterministic IDS algorithm, which is more computationally expensive than our algorithm since our LinIMED algorithms are fully deterministic (the support of the allocation in Kirschner et al. (2021) is two). IDS defines a more complicated version of information gain to achieve asymptotically optimality. Finally, to the best of our knowledge, all these IDS algorithms are designed for linear bandits under the setting that the arm set is fixed and finite, while in our setting we assume the arm set is finite and can change over time. We discuss comparisons to other related work in Appendix B.

## 4 Theorem Statements

**Theorem 1.** *Under Assumption 1, the assumption that $\langle\theta^*, x_{t,a}\rangle \geq 0$ for all $t \geq 1$ and $a \in \mathcal{A}_t$, and the assumption that $\sqrt{\lambda}S \geq 1$, the regret of the LinIMED-1 algorithm is upper bounded as follows:*

$$R_T \leq O\big(d\sqrt{T}\log^{\frac{3}{2}}(T)\big).$$

A proof sketch of Theorem 1 is provided in Section 5.

**Theorem 2.** *Under Assumption 1, and the assumption that $\sqrt{\lambda}S \geq 1$, the regret of the LinIMED-2 algorithm is upper bounded as follows:*

$$R_T \leq O\big(d\sqrt{T}\log^{\frac{3}{2}}(T)\big).$$

**Theorem 3.** *Under Assumption 1, the assumption that $\sqrt{\lambda}S \geq 1$, and that $C$ in Line 11 is a constant, the regret of the LinIMED-3 algorithm is upper bounded as follows:*

$$R_T \leq O\big(d\sqrt{T}\log(T)\big).$$

**Theorem 4.** *Under Assumption 1, the assumption that $L = S = 1$, the regret of the SupLinIMED algorithm (which is applicable to linear bandit problems with $K < \infty$ arms) is upper bounded as follows:*

$$R_T \leq O\bigg(\sqrt{dT\log^3(KT)}\bigg).$$

The upper bounds on the regret of LinIMED and its variants are all of the form $\widetilde{O}(d\sqrt{T})$, which, ignoring the logarithmic term, is the same as OFUL algorithm (Abbasi-Yadkori et al. (2011)). Compared to LinTS, it has an advantage of $O(\sqrt{d} \wedge \sqrt{\log K})$. Also, these upper bounds do not depend on the number of arms $K$, which means it can be applied to linear bandit problems with a large arm set (including infinite arm sets). One observes that LinIMED-2 and LinIMED-3 do not require the additional assumption that $\langle \theta^*, x_{t,a} \rangle \geq 0$ for all $t \geq 1$ and $a \in \mathcal{A}_t$ to achieve the $\widetilde{O}(d\sqrt{T})$ upper regret bound. It is difficult to prove the regret bound for the LinIMED-1 algorithm without this assumption since in our proof we need to use that $\langle \theta^*, X_t \rangle \geq 0$ for any time $t$ to bound the $F_1$ term. On the other hand, LinIMED-2 and LinIMED-3 encourage more exploitations in terms of the index of the empirically best arm at each time without adversely influencing the regret bound; this will accelerate the learning with well-preprocessed datasets. The regret bound of LinIMED-3, in fact, matches that of LinUCB with OFUL's confidence bound. In the proof, we will extensively use a technique known as the "peeling device" (Lattimore & Szepesvári, 2020, Chapter 9). This analytical technique, commonly used in the theory of bandit algorithms, involves the partitioning of the range of some random variable into several pieces, then using the basic fact that $\mathbb{P}(A \cap (\cup_{i=1}^{\infty} B_i)) \leq \sum_{i=1}^{\infty} \mathbb{P}(A \cap B_i)$, we can utilize the more *refined range* of the random variable to derive desired bounds.

Finally, Theorem 4 says that when the arm set is finite, we can use the framework of SupLinUCB (Chu et al., 2011) with our LinIMED index $I_{t,a}$ to achieve a regret bound of the order of $\widetilde{O}(\sqrt{dT})$, which is $\sqrt{d}$ better than the regret bounds yielded by the family of LinIMED algorithms (ignoring the logarithmic terms). The proof is provided in Appendix F.

## 5 Proof Sketch of Theorem 1

We choose to present the proof sketch of Theorem 1 since it contains the main ingredients for all the theorems in the preceding section. Before presenting the proof, we introduce the following lemma and corollary.

**Lemma 1.** *(Abbasi-Yadkori et al. (2011, Theorem 2)) Under Assumption 1, for any time step $t \geq 1$ and any $\gamma > 0$, we have*

$$\mathbb{P}\big(\|\hat{\theta}_{t-1} - \theta^*\|_{V_{t-1}} \leq \sqrt{\beta_{t-1}(\gamma)}\big) \geq 1 - \gamma.$$

This lemma illustrates that the true parameter $\theta^*$ lies in the ellipsoid centered at $\hat{\theta}_{t-1}$ with high probability, which also states the width of the confidence bound.

The second is a corollary of the elliptical potential count lemma in Abbasi-Yadkori et al. (2011):

**Corollary 1.** *(Corollary of Lattimore & Szepesvári (2020, Exercise 19.3)) Assume that $V_0 = \lambda I$ and $\|X_t\| \leq L$ for $t \in [T]$, for any constant $0 < m \leq 2$, the following holds:*

$$\sum_{t=1}^{T} \mathbb{1}\left\{\|X_t\|_{V_{t-1}^{-1}}^2 \geq m\right\} \leq \frac{6d}{m}\log\left(1 + \frac{2L^2}{\lambda m}\right).$$

We remark that this lemma is slightly stronger than the classical elliptical potential lemma since it reveals information about the upper bound of frequency that $\|X_t\|^2_{V^{-1}_{t-1}}$ exceeds some value $m$. Equipped with this lemma, we can perform the peeling device on $\|X_t\|^2_{V^{-1}_{t-1}}$ in our proof of the regret bound, which is a novel technique to the best of our knowledge.

*Proof.* First we define $a^*_t$ as the best arm in time step $t$ such that $a^*_t = \arg\max_{a \in \mathcal{A}_t} \langle \theta^*, x_{t,a} \rangle$, and use $x^*_t := x_{t,a^*_t}$ denote its corresponding context. Let $\Delta_t := \langle \theta^*, x^*_t \rangle - \langle \theta^*, X_t \rangle$ denote the regret in time $t$. Define the following events:

$$B_t := \big\{ \|\hat{\theta}_{t-1} - \theta^*\|_{V_{t-1}} \le \sqrt{\beta_{t-1}(\gamma)} \big\}, \quad C_t := \big\{ \max_{b \in \mathcal{A}_t} \langle \hat{\theta}_{t-1}, x_{t,b} \rangle > \langle \theta^*, x^*_t \rangle - \delta \big\}, \quad D_t := \big\{ \hat{\Delta}_{t,A_t} \ge \varepsilon \big\}.$$

where $\delta$ and $\varepsilon$ are free parameters set to be $\delta = \frac{\Delta_t}{\sqrt{\log T}}$ and $\varepsilon = (1 - \frac{2}{\sqrt{\log T}})\Delta_t$ in this proof sketch.

Then the expected regret $R_T = \mathbb{E} \sum_{t=1}^T \Delta_t$ can be partitioned by events $B_t, C_t, D_t$ such that:

$$R_T = \underbrace{\mathbb{E} \sum_{t=1}^T \Delta_t \cdot \mathbb{1}\{B_t, C_t, D_t\}}_{=:F_1} + \underbrace{\mathbb{E} \sum_{t=1}^T \Delta_t \cdot \mathbb{1}\{B_t, C_t, \overline{D_t}\}}_{=:F_2} + \underbrace{\mathbb{E} \sum_{t=1}^T \Delta_t \cdot \mathbb{1}\{B_t, \overline{C_t}\}}_{=:F_3} + \underbrace{\mathbb{E} \sum_{t=1}^T \Delta_t \cdot \mathbb{1}\{\overline{B_t}\}}_{=:F_4}.$$

For $F_1$, from the event $C_t$ and the fact that $\langle \theta^*, x^*_t \rangle = \Delta_t + \langle \theta^*, X_t \rangle \ge \Delta_t$ (here is where we use that $\langle \theta^*, x_{t,a} \rangle \ge 0$ for all $t$ and $a$), we obtain $\max_{b \in \mathcal{A}_t} \langle \hat{\theta}_{t-1}, x_{t,b} \rangle > (1 - \frac{1}{\sqrt{\log T}})\Delta_t$. For convenience, define $\hat{A}_t := \arg\max_{b \in \mathcal{A}_t} \langle \hat{\theta}_{t-1}, x_{t,b} \rangle$ as the empirically best arm at time step $t$, where ties are broken arbitrarily, then use $\hat{X}_t$ to denote the corresponding context of the arm $\hat{A}_t$. Therefore from the Cauchy–Schwarz inequality, we have $\|\hat{\theta}_{t-1}\|_{V_{t-1}} \|\hat{X}_t\|_{V^{-1}_{t-1}} \ge \langle \hat{\theta}_{t-1}, \hat{X}_t \rangle > (1 - \frac{1}{\sqrt{\log T}})\Delta_t$. This implies that

$$\|\hat{X}_t\|_{V^{-1}_{t-1}} \ge \frac{(1 - \frac{1}{\sqrt{\log T}})\Delta_t}{\|\hat{\theta}_{t-1}\|_{V_{t-1}}} \ . \tag{2}$$

On the other hand, we claim that $\|\hat{\theta}_{t-1}\|_{V_{t-1}}$ can be upper bounded as $O(\sqrt{T})$. This can be seen from the fact that $\|\hat{\theta}_{t-1}\|_{V_{t-1}} = \|\hat{\theta}_{t-1} - \theta^* + \theta^*\|_{V_{t-1}} \le \|\hat{\theta}_{t-1} - \theta^*\|_{V_{t-1}} + \|\theta^*\|_{V_{t-1}}$. Since the event $B_t$ holds, we know the first term is upper bounded by $\sqrt{\beta_{t-1}(\gamma)}$, and since the largest eigenvalue of the matrix $V_{t-1}$ is upper bounded by $\lambda + TL^2$ and $\|\theta^*\| \le S$, the second term is upper bounded by $S\sqrt{\lambda + TL^2}$. Hence, $\|\hat{\theta}_{t-1}\|_{V_{t-1}}$ is upper bounded by $O(\sqrt{T})$. Then one can substitute this bound back into Eqn. (2), and this yields

$$\|\hat{X}_t\|_{V^{-1}_{t-1}} \ge \Omega\Big(\frac{1}{\sqrt{T}}\Big(1 - \frac{1}{\sqrt{\log T}}\Big)\Delta_t\Big) \ . \tag{3}$$

Furthermore, by our design of the algorithm, the index of $A_t$ is not larger than the index of the arm with the largest empirical reward at time $t$. Hence,

$$I_{t,A_t} = \frac{\hat{\Delta}^2_{t,A_t}}{\beta_{t-1}(\gamma)\|X_t\|^2_{V^{-1}_{t-1}}} + \log \frac{1}{\beta_{t-1}(\gamma)\|X_t\|^2_{V^{-1}_{t-1}}} \le \log \frac{1}{\beta_{t-1}(\gamma)\|\hat{X}_t\|^2_{V^{-1}_{t-1}}} \ . \tag{4}$$

In the following, we set $\gamma$ as well as another free parameter $\Gamma$ as follows:

$$\Gamma = \frac{d \log^{\frac{3}{2}} T}{\sqrt{T}} \quad \text{and} \quad \gamma = \frac{1}{t^2}, . \tag{5}$$

If $\|X_t\|^2_{V^{-1}_{t-1}} \ge \frac{\Delta^2_t}{\beta_{t-1}(\gamma)}$, by using Corollary 1 with the choice in Eqn. (5), the upper bound of $F_1$ in this case is $O(d\sqrt{T \log T})$. Otherwise, using the event $D_t$ and the bound in (3), we deduce that for all $T$ sufficiently

large, we have $\|X_t\|^2_{V^{-1}_{t-1}} \geq \Omega\left(\frac{\Delta^2_t}{\beta_{t-1}(\gamma)\log(T/\Delta^2_t)}\right)$. Therefore by using Corollary 1 and the "peeling device" (Lattimore & Szepesvári, 2020, Chapter 9) on $\Delta_t$ such that $2^{-l} < \Delta_t \leq 2^{-l+1}$ for $l = 1, 2, \ldots, \lceil Q \rceil$ where $Q = -\log_2 \Gamma$ and $\Gamma$ is chosen as in Eqn. (5). Now consider,

$$
\begin{aligned}
F_1 &\leq O(1) + \mathbb{E}\sum^T_{t=1} \Delta_t \cdot \mathbb{1}\left\{\|X_t\|^2_{V^{-1}_{t-1}} \geq \Omega\left(\frac{\Delta^2_t}{\beta_{t-1}(\gamma)\log(T/\Delta^2_t)}\right)\right\} \\
&\leq O(1) + T\Gamma + \mathbb{E}\sum^T_{t=1}\sum^{\lceil Q \rceil}_{l=1} \Delta_t \cdot \mathbb{1}\left\{\|X_t\|^2_{V^{-1}_{t-1}} \geq \Omega\left(\frac{\Delta^2_t}{\beta_{t-1}(\gamma)\log(T/\Delta^2_t)}\right)\right\}\mathbb{1}\left\{2^{-l} < \Delta_t \leq 2^{-l+1}\right\} \\
&\leq O(1) + T\Gamma + \mathbb{E}\sum^T_{t=1}\sum^{\lceil Q \rceil}_{l=1} 2^{-l+1} \cdot \mathbb{1}\left\{\|X_t\|^2_{V^{-1}_{t-1}} \geq \Omega\left(\frac{2^{-2l}}{\beta_{t-1}(\gamma)\log(T \cdot 2^{2l})}\right)\right\} \\
&\leq O(1) + T\Gamma + \mathbb{E}\sum^{\lceil Q \rceil}_{l=1} 2^{-l+1}O\left(2^{2l}d\beta_T(\gamma)\log(2^{2l}T)\log\left(1 + \frac{2L^2 \cdot 2^{2l}\beta_T(\gamma)\log(T \cdot 2^{2l})}{\lambda}\right)\right) \quad (6) \\
&\leq O(1) + T\Gamma + \sum^{\lceil Q \rceil}_{l=1} 2^{l+1} \cdot O\left(d\beta_T(\gamma)\log(\frac{T}{\Gamma^2})\log\left(1 + \frac{L^2\beta_T(\gamma)\log(\frac{T}{\Gamma^2})}{\lambda\Gamma^2}\right)\right) \\
&\leq O(1) + T\Gamma + O\left(\frac{d\beta_T(\gamma)\log(\frac{T}{\Gamma^2})}{\Gamma}\log\left(1 + \frac{L^2\beta_T(\gamma)\log(\frac{T}{\Gamma^2})}{\lambda\Gamma^2}\right)\right), \quad (7)
\end{aligned}
$$

where in Inequality (6) we used Corollary 1. Substituting the choices of $\Gamma$ and $\gamma$ in (5) into (7) yields the upper bound on $\mathbb{E}\sum^T_{t=1} \Delta_t \cdot \mathbb{1}\{B_t, C_t, D_t\} \cdot \mathbb{1}\{\|X_t\|^2_{V^{-1}_{t-1}} < \frac{\Delta^2_t}{\beta_{t-1}(\gamma)}\}$ of the order $O(d\sqrt{T}\log^{\frac{3}{2}} T)$. Hence $F_1 \leq O(d\sqrt{T}\log^{\frac{3}{2}} T)$. Other details are fleshed out in Appendix C.2.

For $F_2$, since $C_t$ and $\overline{D}_t$ together imply that $\langle \theta^*, x^*_t \rangle - \delta < \varepsilon + \langle \hat{\theta}_{t-1}, X_t \rangle$, then using the choices of $\delta$ and $\varepsilon$, we have $\langle \hat{\theta}_{t-1} - \theta^*, X_t \rangle > \frac{\Delta_t}{\sqrt{\log T}}$. Substituting this into the event $B_t$ and using the Cauchy–Schwarz inequality, we have

$$
\|X_t\|^2_{V^{-1}_{t-1}} \geq \frac{\Delta^2_t}{\beta_{t-1}(\gamma)\log T}.
$$

Again applying the "peeling device" on $\Delta_t$ and Corollary 1, we can upper bound $F_2$ as follows:

$$
F_2 \leq T\Gamma + O\left(\frac{d\beta_T(\gamma)\log T}{\Gamma}\right)\log\left(1 + \frac{L^2\beta_T(\gamma)\log T}{\lambda\Gamma^2}\right). \quad (8)
$$

Then with the choice of $\Gamma$ and $\gamma$ as stated in (5), the upper bound of the $F_2$ is also of order $O(d\sqrt{T}\log^{\frac{3}{2}} T)$. More details of the calculation leading to Eqn. (8) are in Appendix C.3.

For $F_3$, this is the case when the best arm at time $t$ does not perform sufficiently well so that the empirically largest reward at time $t$ is far from the highest expected reward. One observes that minimizing $F_3$ results in a tradeoff with respect to $F_1$. On the event $\overline{C}_t$, we can again apply the "peeling device" on $\langle \theta^*, x^*_t \rangle - \langle \hat{\theta}_{t-1}, x^*_t \rangle$ such that $\frac{q+1}{2}\delta \leq \langle \theta^*, x^*_t \rangle - \langle \hat{\theta}_{t-1}, x^*_t \rangle < \frac{q+2}{2}\delta$ where $q \in \mathbb{N}$. Then using the fact that $I_{t,A_t} \leq I_{t,a^*_t}$, we have

$$
\log \frac{1}{\beta_{t-1}(\gamma)\|X_t\|^2_{V^{-1}_{t-1}}} < \frac{q^2\delta^2}{4\beta_{t-1}(\gamma)\|x^*_t\|^2_{V^{-1}_{t-1}}} + \log\frac{1}{\beta_{t-1}(\gamma)\|x^*_t\|^2_{V^{-1}_{t-1}}}. \quad (9)
$$

On the other hand, using the event $B_t$ and the Cauchy–Schwarz inequality, it holds that

$$
\|x^*_t\|_{V^{-1}_{t-1}} \geq \frac{(q+1)\delta}{2\sqrt{\beta_{t-1}(\gamma)}}. \quad (10)
$$

If $\|X_t\|^2_{V^{-1}_{t-1}} \geq \frac{\Delta^2_t}{\beta_{t-1}(\gamma)}$, the regret in this case is bounded by $O(d\sqrt{T \log T})$. Otherwise, combining Eqn. (9) and Eqn. (10) implies that

$$\|X_t\|^2_{V^{-1}_{t-1}} \geq \frac{(q+1)^2\delta^2}{4\beta_{t-1}(\gamma)} \exp\left(-\frac{q^2}{(q+1)^2}\right).$$

Using Corollary 1, we can now conclude that $F_3$ is upper bounded as

$$F_3 \leq T\Gamma + O\left(\frac{d\beta_T(\gamma)\log T}{\Gamma}\right) \log\left(1 + \frac{L^2\beta_T(\gamma)\log T}{\lambda\Gamma^2}\right). \tag{11}$$

Substituting $\Gamma$ and $\gamma$ in Eqn. (5) into Eqn. (11), we can upper bound $F_3$ by $O(d\sqrt{T}\log^{\frac{3}{2}} T)$. Complete details are provided in Appendix C.4.

For $F_4$, using Lemma 1 with the choice of $\gamma = 1/t^2$ and $Q = -\log\Gamma$, we have

$$F_4 = \mathbb{E}\sum_{t=1}^{T}\Delta_t \cdot \mathbb{1}\left\{\overline{B}_t\right\} \leq T\Gamma + \mathbb{E}\sum_{t=1}^{T}\sum_{l=1}^{\lceil Q\rceil}\Delta_t \cdot \mathbb{1}\left\{2^{-l} < \Delta_t \leq 2^{-l+1}\right\}\mathbb{1}\left\{\overline{B}_t\right\}$$

$$\leq T\Gamma + \sum_{t=1}^{T}\sum_{l=1}^{\lceil Q\rceil}2^{-l+1}\mathbb{P}\left(\overline{B}_t\right) \leq T\Gamma + \sum_{t=1}^{T}\sum_{l=1}^{\lceil Q\rceil}2^{-l+1}\gamma < T\Gamma + \frac{\pi^2}{3}.$$

Thus, $F_4 \leq O(d\sqrt{T}\log^{\frac{3}{2}} T)$. In conclusion, with the choice of $\Gamma$ and $\gamma$ in Eqn. (5), we have shown that the expected regret of LinIMED-1 $R_T = \sum_{i=1}^{4} F_i$ is upper bounded by $O(d\sqrt{T}\log^{\frac{3}{2}} T)$. $\qquad\square$

For LinIMED-2, the proof is similar but the assumption that $\langle\theta^*, x_{t,a}\rangle \geq 0$ is not required. For LinIMED-3, by directly using the UCB in Line 6 of Algorithm 1, we improve the regret bound to match the state-of-the-art $O(d\sqrt{T}\log T)$, which matches that of LinUCB with OFUL's confidence bound.

## 6 Empirical Studies

This section aims to justify the utility of the family of LinIMED algorithms we developed and to demonstrate their effectiveness through quantitative evaluations in simulated environments and real-world datasets such as the MovieLens dataset. We compare our LinIMED algorithms with LinTS and LinUCB with the choice $\lambda = L^2$. We set $\beta_t(\gamma) = (R\sqrt{3d\log(1+t)} + \sqrt{2})^2$ (here $\gamma = \frac{1}{(1+t)^2}$ and $L = \sqrt{2}$) for the synthetic dataset with varying and finite arm set and $\beta_t(\gamma) = (R\sqrt{d\log((1+t)t^2)} + \sqrt{20})^2$ (here $\gamma = \frac{1}{t^2}$ and $L = \sqrt{20}$) for the MovieLens dataset respectively. The confidence widths $\sqrt{\beta_t(\gamma)}$ for each algorithm are multiplied by a factor $\alpha$ and we tune $\alpha$ by searching over the grid $\{0.05, 0.1, 0.15, 0.2, \ldots, 0.95, 1.0\}$ and report the **best performance** for each algorithm; see Appendix G. Both $\gamma$'s are of order $O(\frac{1}{t^2})$ as suggested by our proof sketch in Eqn. (5). We set $C = 30$ in LinIMED-3 throughout. The sub-Gaussian noise level is $R = 0.1$. We choose LinUCB and LinTS as competing algorithms since they are paradigmatic examples of deterministic and randomized contextual linear bandit algorithms respectively. We also included IDS in our comparisons for the fixed and finite arm set settings. Finally, we only show the performances of SupLinUCB and SupLinIMED algorithms but only in Figs. 1 and 2 since it is well known that there is a substantial performance degradation compared to established methodologies like LinUCB or LinTS (as mentioned in Lattimore & Szepesvári (2020, Chapter 22) and also seen in Figs. 1 and 2.

### 6.1 Experiments on a Synthetic Dataset in the Varying Arm Set Setting

We perform an empirical study on a varying arm setting. We evaluate the performance with different dimensions $d$ and different number of arms $K$. We set the unknown parameter vector and the best context vector as $\theta^* = x^*_t = [\frac{1}{\sqrt{d-1}}, \ldots, \frac{1}{\sqrt{d-1}}, 0]^\top \in \mathbb{R}^d$. There are $K-2$ suboptimal arms vectors, which are all the same (i.e., repeated) and share the context $[(1-\frac{1}{7+z_{t,i}})\frac{1}{\sqrt{d-1}}, \ldots, (1-\frac{1}{7+z_{t,i}})\frac{1}{\sqrt{d-1}}, (1-\frac{1}{7+z_{t,i}})]^\top \in \mathbb{R}^d$

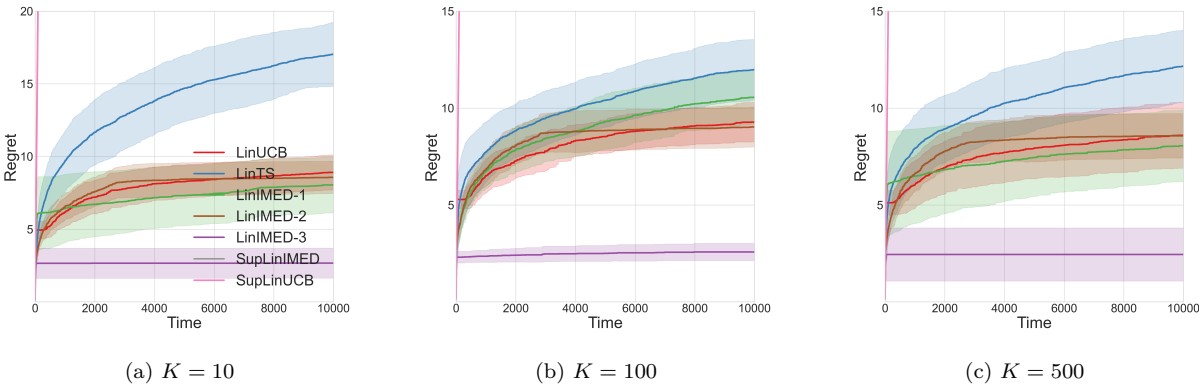

(a) $K = 10$  (b) $K = 100$  (c) $K = 500$

Figure 1: Simulation results (expected regrets) on the synthetic dataset with different $K$'s

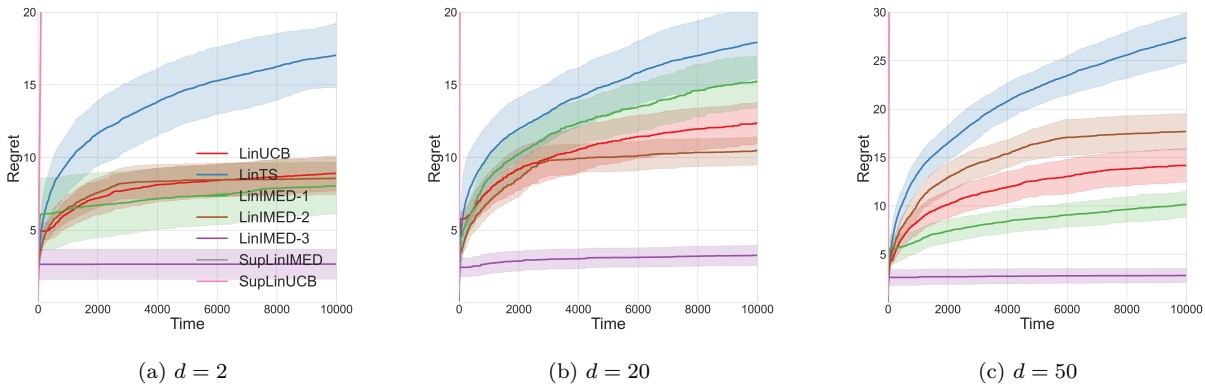

(a) $d = 2$  (b) $d = 20$  (c) $d = 50$

Figure 2: Simulation results (expected regrets) on the synthetic dataset with different $d$'s

where $z_{t,i} \sim \text{Uniform}[0, 0.1]$ is iid noise for each $t \in [T]$ and $i \in [K - 2]$. Finally, there is also one "worst" arm vector with context $[0, 0, \ldots, 0, 1]^\top$. First we fix $d = 2$. The results for different numbers of arms such as $K = 10, 100, 500$ are shown in Fig. 1. Note that each plot is repeated 50 times to obtain the mean and standard deviation of the regret. From Fig. 1, we observe that LinIMED-1 and LinIMED-2 are comparable to LinUCB and LinTS, while LinIMED-3 outperforms LinTS and LinUCB regardless of the number of the arms $K$. Second, we set $K = 10$ with the dimensions $d = 2, 20, 50$. Each trial is again repeated 50 times and the regret over time is shown in Fig. 2. Again, we see that LinIMED-1 and LinIMED-2 are comparable to LinUCB and LinTS, LinIMED-3 clearly perform better than LinUCB and LinTS.

The experimental results on synthetic data demonstrate that the performances of LinIMED-1 and LinIMED-2 are largely similar but LinIMED-3 is slightly superior (corroborating our theoretical findings). More importantly, LinIMED-3 outperforms both the LinTS and LinUCB algorithms in a statistically significant manner, regardless of the number of arms $K$ or the dimension $d$ of the data.

## 6.2 Experiments on the "End of Optimism" instance

Algorithms based on the optimism principle such as LinUCB and LinTS have been shown to be not asymptotically optimal. A paradigmatic example is known as the "End of Optimism" (Lattimore & Szepesvari, 2017; Kirschner et al., 2021)). In this two-dimensional case in which the true parameter vector $\theta^* = [1; 0]$, there are three arms represented by the arm vectors: $[1; 0], [0; 1]$ and $[1 - \varepsilon; 2\varepsilon]$, where $\varepsilon > 0$ is small. In this example, it is observed that even pulling a highly suboptimal arm (the second one) provides a lot of information about the best arm (the first one). We perform experiments with the same confidence parameter $\beta_t(\gamma) = (R\sqrt{3d \log(1 + t)} + \sqrt{2})^2$ as in Section 6.1 (where the noise level $R = 0.1$, dimension $d = 2$). We also include the asymptotically optimal IDS algorithm (Kirschner et al. (2021) with the choice of $\eta_s = \beta_s(\delta)^{-1}$;

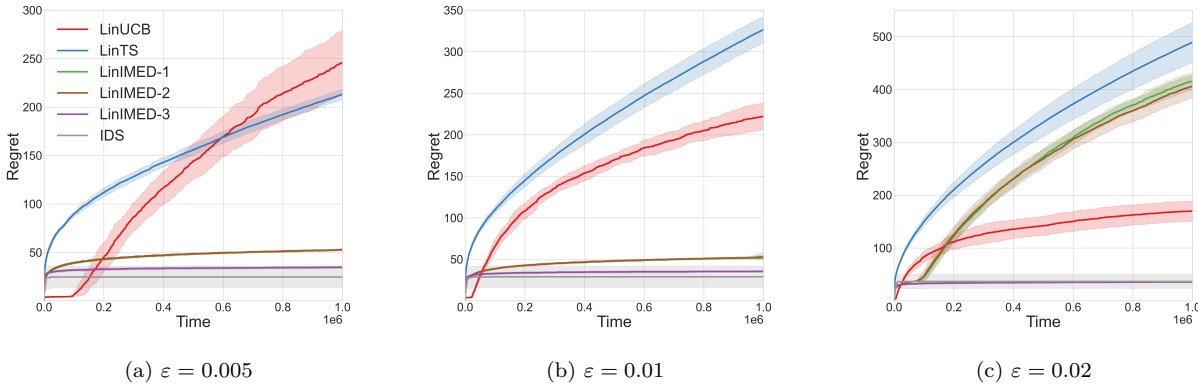

(a) $\varepsilon = 0.005$        (b) $\varepsilon = 0.01$        (c) $\varepsilon = 0.02$

Figure 3: Simulation results (expected regrets) on the "End of Optimism" instance with different $\varepsilon$'s

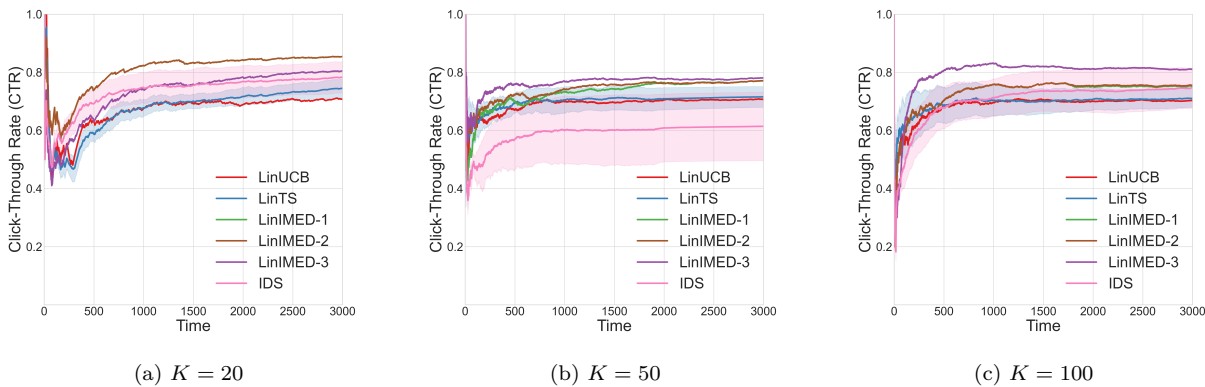

(a) $K = 20$        (b) $K = 50$        (c) $K = 100$

Figure 4: Simulation results (CTRs) of the MovieLens dataset with different $K$'s

this is suggested in Kirschner et al. (2021). Each algorithm is run over 10 independent trials. The regrets of all competing algorithms are shown in Fig. 3 with $\varepsilon = 0.05, 0.01, 0.02$ and for a fixed horizon $T = 10^6$.

From Fig. 3 we observe that the LinIMED algorithms perform much better than LinUCB and LinTS and LinIMED-3 is comparable to IDS in this "End of Optimism" instance. In particular, LinIMED-3 performs significantly better than LinUCB and LinTS even when $\varepsilon$ is of a moderate value such as $\varepsilon = 0.02$. We surmise that the reason behind the superior performance of our LinIMED algorithms on the "End of Optimism" instance is that the first term of our LinIMED index is $\hat{\Delta}_{t,a}^2/(\beta_{t-1}(\gamma)\|x_{t,a}\|_{V_{t-1}^{-1}}^2)$, which can be viewed as an approximate and simpler version of the information ratio that movtivates the design the IDS) algorithm.

### 6.3 Experiments on the MovieLens Dataset

The MovieLens dataset (Cantador et al. (2011)) is a widely-used benchmark dataset for research in recommendation systems. We specifically choose to use the MovieLens 10M dataset, which contains 10 million ratings (from 0 to 5) and 100,000 tag applications applied to 10,000 movies by 72,000 users. To preprocess the dataset, we choose the best $K \in \{20, 50, 100\}$ movies for consideration. At each time $t$, one random user visits the website and is recommended one of the best $K$ movies. We assume that the user will click on the recommended movie if and only if the user's rating of this movie is at least 3. We implement the three versions of LinIMED, LinUCB, LinTS and IDS on this dataset. Each trial is repeated over 100 runs and the averages and standard deviations of the click-through rates (CTRs) as functions of time are reported in Fig. 4. One observes that LinIMED variants significantly outperform LinUCB and LinTS for all $K \in \{20, 50, 100\}$ when

time horizon $T$ is sufficiently large. LinIMED-1 and LinIMED-2 perform significantly better than IDS when $k = 20, 50$, LinIMED-3 perform significantly better than IDS when $k = 50, 100$. Furthermore, by virtue of the fact that IDS is randomized, the variance of IDS is higher than that of LinIMED.

## 7 Future Work

In the future, a fruitful direction of research is to further modify the LinIMED algorithm to make it also asymptotically optimal; we believe that in this case, the analysis would be more challenging, but the theoretical and empirical performances might be superior to our three LinIMED algorithms. In addition, one can generalize the family of IMED-style algorithms to generalized linear bandits or neural contextual bandits.

## Acknowledgements

This work is supported by funding from a Ministry of Education Academic Research Fund (AcRF) Tier 2 grant under grant number A-8000423-00-00 and AcRF Tier 1 grants under grant numbers A-8000189-01-00 and A-8000980-00-00. This research is supported by the National Research Foundation, Singapore under its AI Singapore Programme (AISG Award No: AISG2-PhD-2023-08-044T-J), and is part of the programme DesCartes which is supported by the National Research Foundation, Prime Minister's Office, Singapore under its Campus for Research Excellence and Technological Enterprise (CREATE) programme.

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

# Supplementary Materials for the TMLR submission "Linear Indexed Minimum Empirical Divergence Algorithms"

## A  BaseLinUCB Algorithm

Here, we present the BaseLinUCB algorithm used as a subroutine in SubLinIMED (Algorithm 2).

---

**Algorithm 3** BaseLinUCB

---

1: **Input:** $\gamma = \frac{1}{2t^2}$, $\alpha = \sqrt{\frac{1}{2} \ln \frac{2TK}{\gamma}}$, $\Psi_t \subseteq \{1, 2, ..., t-1\}$
2: $V_t = I_d + \sum_{\tau \in \Psi_t} x_{\tau, A_\tau}^T x_{\tau, A_\tau}$
3: $b_t = \sum_{\tau \in \Psi_t} Y_{\tau, A_\tau} x_{\tau, A_\tau}$
4: $\hat{\theta}_t = V_t^{-1} b_t$
5: Observe $K$ arm features $x_{t,1}, x_{t,2}, \ldots, x_{t,K} \in \mathbb{R}^d$
6: **for** $a \in [K]$ **do**
7:      $w_{t,a} = \alpha \sqrt{x_{t,a}^T V_t^{-1} x_{t,a}}$
8:      $\hat{Y}_{t,a} = \langle \hat{\theta}_t, x_{t,a} \rangle$
9: **end for**

---

## B  Comparison to other related work

Saber et al. (2021) adopts the IMED algorithm to unimodal bandits which achieves asymptotically optimality for one-dimensional exponential family distributions. In their algorithm IMED-UB, they narrow down the search region to the neighboring regions of the empirically best arm and then implement the IMED algorithm for $K$-armed bandit as in Honda & Takemura (2015). This design is inspired by the lower bound and only involves the neighboring arms of the best arm. The settings in which the algorithm in Saber et al. (2021) is applied to is different from our proposed LinIMED algorithms as we focus on linear bandits, not unimodal bandits.

Liu et al. (2024) proposes an algorithm that achieves $\widetilde{O}(\sqrt{T})$ regret for adversarial linear bandits with stochastic action sets in the absence of a simulator or prior knowledge on the distribution. Although their setting is different from ours, they also use a bonus term $-\alpha_t \hat{\Sigma}_t^{-1}$ in the lifted covariance matrix to encourage exploration. This is similar to our choice of the second term $\log(1/\beta_{t-1}(\gamma) \|x_{t,a}\|_{V_{t-1}^{-1}}^2)$ in LinIMED-1.

## C  Proof of the regret bound for LinIMED-1 (Complete proof of Theorem 1)

Here and in the following, we abbreviate $\beta_t(\gamma)$ as $\beta_t$, i.e., we drop the dependence of $\beta_t$ on $\gamma$, which is taken to be $\frac{1}{t^2}$ per Eqn. (5).

### C.1  Statement of Lemmas for LinIMED-1

We first state the following lemmas which respectively show the upper bound of $F_1$ to $F_4$:

**Lemma 2.** *Under Assumption 1, the assumption that $\langle \theta^*, x_{t,a} \rangle \geq 0$ for all $t \geq 1$ and $a \in \mathcal{A}_t$, and the assumption that $\sqrt{\lambda} S \geq 1$, then for the free parameter $0 < \Gamma < 1$, the term $F_1$ for LinIMED-1 satisfies:*

$$F_1 \leq O(1) + T\Gamma + O\left( \frac{d\beta_T \log(\frac{T}{\Gamma^2})}{\Gamma} \log\left( 1 + \frac{L^2 \beta_T \log(\frac{T}{\Gamma^2})}{\lambda \Gamma^2} \right) \right). \tag{12}$$

*With the choice of $\Gamma$ as in Eqn. (5),*

$$F_1 \leq O\left( d\sqrt{T} \log^{\frac{3}{2}} T \right).$$

---

**Lemma 3.** *Under Assumption 1, and the assumption that $\sqrt{\lambda}S \geq 1$, for the free parameter $0 < \Gamma < 1$, the term $F_2$ for LinIMED-1 satisfies:*

$$F_2 \leq 2T\Gamma + O\left(\frac{d\beta_T \log T}{\Gamma}\right) \log\left(1 + \frac{L^2\beta_T \log T}{\lambda\Gamma^2}\right) . \tag{13}$$

*With the choice of $\Gamma$ as in Eqn. (5),*

$$F_2 \leq O\left(d\sqrt{T} \log^{\frac{3}{2}} T\right) .$$

**Lemma 4.** *Under Assumption 1, and the assumption that $\sqrt{\lambda}S \geq 1$, for the free parameter $0 < \Gamma < 1$, the term $F_3$ for LinIMED-1 satisfies:*

$$F_3 \leq 2T\Gamma + O\left(\frac{d\beta_T \log(T)}{\Gamma} \log\left(1 + \frac{L^2\beta_T \log(T)}{\lambda\Gamma^2}\right)\right) . \tag{14}$$

*With the choice of $\Gamma$ as in Eqn. (5),*

$$F_3 \leq O\left(d\sqrt{T} \log^{\frac{3}{2}} T\right) .$$

**Lemma 5.** *Under Assumption 1, for the free parameter $0 < \Gamma < 1$, the term $F_4$ for LinIMED-1 satisfies:*

$$F_4 \leq T\Gamma + O(1) .$$

*With the choice of $\Gamma$ as in Eqn. (5),*

$$F_4 \leq O\left(d\sqrt{T} \log^{\frac{3}{2}} T\right) .$$

## C.2  Proof of Lemma 2

*Proof.* From the event $C_t$ and the fact that $\langle \theta^*, x_t^* \rangle = \Delta_t + \langle \theta^*, X_t \rangle \geq \Delta_t$ (here is where we use that $\langle \theta^*, x_{t,a} \rangle \geq 0$ for all $t$ and $a$), we obtain $\max_{b \in \mathcal{A}_t} \langle \hat{\theta}_{t-1}, x_{t,b} \rangle > (1 - \frac{1}{\sqrt{\log T}})\Delta_t$. For convenience, define $\hat{A}_t := \arg\max_{b \in \mathcal{A}_t} \langle \hat{\theta}_{t-1}, x_{t,b} \rangle$ as the empirically best arm at time step $t$, where ties are broken arbitrarily, then use $\hat{X}_t$ to denote the corresponding context of the arm $\hat{A}_t$. Therefore from the Cauchy–Schwarz inequality, we have $\|\hat{\theta}_{t-1}\|_{V_{t-1}}\|\hat{X}_t\|_{V_{t-1}^{-1}} \geq \langle \hat{\theta}_{t-1}, \hat{X}_t \rangle > (1 - \frac{1}{\sqrt{\log T}})\Delta_t$. This implies that

$$\|\hat{X}_t\|_{V_{t-1}^{-1}} \geq \frac{(1 - \frac{1}{\sqrt{\log T}})\Delta_t}{\|\hat{\theta}_{t-1}\|_{V_{t-1}}} .$$

On the other hand, we claim that $\|\hat{\theta}_{t-1}\|_{V_{t-1}}$ can be upper bounded as $O(\sqrt{T})$. This can be seen from the fact that $\|\hat{\theta}_{t-1}\|_{V_{t-1}} = \|\hat{\theta}_{t-1} - \theta^* + \theta^*\|_{V_{t-1}} \leq \|\hat{\theta}_{t-1} - \theta^*\|_{V_{t-1}} + \|\theta^*\|_{V_{t-1}}$. Since the event $B_t$ holds, we know the first term is upper bounded by $\sqrt{\beta_{t-1}(\gamma)}$, and since the maximum eigenvalue of the matrix $V_{t-1}$ is upper bounded by $\lambda + TL^2$ and $\|\theta^*\| \leq S$, the second term is upper bounded by $S\sqrt{\lambda + TL^2}$. Hence, $\|\hat{\theta}_{t-1}\|_{V_{t-1}}$ is upper bounded by $O(\sqrt{T})$. Then one can substitute this bound back into Eqn. (2), and this yields

$$\|\hat{X}_t\|_{V_{t-1}^{-1}} \geq \Omega\left(\frac{1}{\sqrt{T}}\left(1 - \frac{1}{\sqrt{\log T}}\right)\Delta_t\right) .$$

Furthermore, by our design of the algorithm, the index of $A_t$ is not larger than the index of the arm with the largest empirical reward at time $t$. Hence,

$$I_{t,A_t} = \frac{\hat{\Delta}_{t,A_t}^2}{\beta_{t-1}(\gamma)\|X_t\|_{V_{t-1}^{-1}}^2} + \log\frac{1}{\beta_{t-1}(\gamma)\|X_t\|_{V_{t-1}^{-1}}^2} \leq \log\frac{1}{\beta_{t-1}(\gamma)\|\hat{X}_t\|_{V_{t-1}^{-1}}^2} .$$

If $\|X_t\|^2_{V^{-1}_{t-1}} \geq \frac{\Delta^2_t}{\beta_{t-1}}$, by using Corollary 1 and the "peeling device" (Lattimore & Szepesvári, 2020, Chapter 9) on $\Delta_t$ such that $2^{-l} < \Delta_t \leq 2^{-l+1}$ for $l = 1, 2, \ldots, \lceil Q \rceil$ where $Q = -\log_2 \Gamma$,

$$\mathbb{E} \sum_{t=1}^{T} \Delta_t \cdot \mathbb{1}\{B_t, C_t, D_t\} \cdot \mathbb{1}\left\{\|X_t\|^2_{V^{-1}_{t-1}} \geq \frac{\Delta^2_t}{\beta_{t-1}}\right\} \leq \mathbb{E} \sum_{t=1}^{T} \Delta_t \cdot \mathbb{1}\left\{\|X_t\|^2_{V^{-1}_{t-1}} \geq \frac{\Delta^2_t}{\beta_{t-1}}\right\} \tag{15}$$

$$= \mathbb{E} \sum_{t=1}^{T} \Delta_t \cdot \mathbb{1}\left\{\|X_t\|^2_{V^{-1}_{t-1}} \geq \frac{\Delta^2_t}{\beta_{t-1}}\right\} \cdot \mathbb{1}\{\Delta_t \leq \Gamma\} + \mathbb{E} \sum_{t=1}^{T} \Delta_t \cdot \mathbb{1}\left\{\|X_t\|^2_{V^{-1}_{t-1}} \geq \frac{\Delta^2_t}{\beta_{t-1}}\right\} \cdot \mathbb{1}\{\Delta_t > \Gamma\}$$

$$\leq T\Gamma + \mathbb{E} \sum_{t=1}^{T} \sum_{l=1}^{\lceil Q \rceil} \Delta_t \cdot \mathbb{1}\left\{\|X_t\|^2_{V^{-1}_{t-1}} \geq \frac{\Delta^2_t}{\beta_{t-1}}\right\} \cdot \mathbb{1}\{2^{-l} < \Delta_t \leq 2^{-l+1}\}$$

$$\leq T\Gamma + \mathbb{E} \sum_{t=1}^{T} \sum_{l=1}^{\lceil Q \rceil} 2^{-l+1} \cdot \mathbb{1}\left\{\|X_t\|^2_{V^{-1}_{t-1}} \geq \frac{2^{-2l}}{\beta_T}\right\}$$

$$\leq T\Gamma + \mathbb{E} \sum_{l=1}^{\lceil Q \rceil} 2^{-l+1} \frac{6d\beta_T}{2^{-2l}} \log\left(1 + \frac{2L^2\beta_T}{\lambda \cdot 2^{-2l}}\right)$$

$$= T\Gamma + \mathbb{E} \sum_{l=1}^{\lceil Q \rceil} 2^l \cdot 12d\beta_T \log\left(1 + \frac{2^{2l+1}L^2\beta_T}{\lambda}\right)$$

$$< T\Gamma + \mathbb{E} \sum_{l=1}^{\lceil Q \rceil} 2^l \cdot 12d\beta_T \log\left(1 + \frac{2^{2Q+3}L^2\beta_T}{\lambda}\right)$$

$$= T\Gamma + (2^{\lceil Q \rceil} - 1) \cdot 24d\beta_T \log\left(1 + \frac{2^{2Q+3}L^2\beta_T}{\lambda}\right)$$

$$< T\Gamma + \frac{48d\beta_T}{\Gamma} \log\left(1 + \frac{8L^2\beta_T}{\lambda\Gamma^2}\right)$$

Then with the choice of $\Gamma$ as in Eqn. (5),

$$\mathbb{E} \sum_{t=1}^{T} \Delta_t \cdot \mathbb{1}\{B_t, C_t, D_t\} \cdot \mathbb{1}\left\{\|X_t\|^2_{V^{-1}_{t-1}} \geq \frac{\Delta^2_t}{\beta_{t-1}}\right\}$$

$$< d\sqrt{T} \log^{\frac{3}{2}} T + \frac{48\beta_T\sqrt{T}}{\log^{\frac{3}{2}} T} \log\left(1 + \frac{8L^2\beta_T T}{\lambda d^2 \log^3 T}\right)$$

$$\leq O\left(d\sqrt{T} \log^{\frac{3}{2}} T\right). \tag{16}$$

Otherwise we have $\|X_t\|^2_{V^{-1}_{t-1}} < \frac{\Delta^2_t}{\beta_{t-1}}$, then $\log \frac{1}{\beta_{t-1}\|X_t\|^2_{V^{-1}_{t-1}}} > 0$ since $\Delta_t \leq 1$. Substituting this into Eqn. (4), then using the event $D_t$ and the bound in (3), we deduce that for all $T$ sufficiently large, we have $\|X_t\|^2_{V^{-1}_{t-1}} \geq \Omega\left(\frac{\Delta^2_t}{\beta_{t-1} \log(T/\Delta^2_t)}\right)$. Therefore by using Corollary 1 and the "peeling device" (Lattimore & Szepesvári, 2020, Chapter 9) on $\Delta_t$ such that $2^{-l} < \Delta_t \leq 2^{-l+1}$ for $l = 1, 2, \ldots, \lceil Q \rceil$ where $\Gamma := 2^{-Q}$ is a free parameter that we can choose. Consider,

$$\mathbb{E} \sum_{t=1}^{T} \Delta_t \cdot \mathbb{1}\{B_t, C_t, D_t\} \cdot \mathbb{1}\left\{\|X_t\|^2_{V^{-1}_{t-1}} < \frac{\Delta^2_t}{\beta_{t-1}}\right\}$$

$$\leq \mathbb{E} \sum_{t=1}^{T} \Delta_t \cdot \mathbb{1}\{B_t, C_t, D_t\} \cdot \mathbb{1}\left\{\Delta_t \leq 2^{-\lceil Q \rceil}\right\} \cdot \mathbb{1}\left\{\|X_t\|^2_{V^{-1}_{t-1}} < \frac{\Delta^2_t}{\beta_{t-1}}\right\}$$

$$+ \mathbb{E} \sum_{t=1}^{T} \Delta_t \cdot \mathbb{1}\{B_t, C_t, D_t\} \cdot \mathbb{1}\left\{\Delta_t > 2^{-\lceil Q \rceil}\right\} \cdot \mathbb{1}\left\{\|X_t\|^2_{V^{-1}_{t-1}} < \frac{\Delta^2_t}{\beta_{t-1}}\right\}$$

$$\leq O(1) + T\Gamma + \mathbb{E}\sum_{t=1}^{T}\Delta_t \cdot \mathbb{1}\left\{\|X_t\|_{V_{t-1}^{-1}}^2 \geq \Omega\Big(\frac{\Delta_t^2}{\beta_{t-1}\log(T/\Delta_t^2)}\Big)\right\}\mathbb{1}\left\{\Delta_t > 2^{-\lceil Q\rceil}\right\}$$

$$\leq O(1) + T\Gamma + \mathbb{E}\sum_{t=1}^{T}\sum_{l=1}^{\lceil Q\rceil}\Delta_t \cdot \mathbb{1}\left\{\|X_t\|_{V_{t-1}^{-1}}^2 \geq \Omega\Big(\frac{\Delta_t^2}{\beta_{t-1}\log(T/\Delta_t^2)}\Big)\right\}\mathbb{1}\left\{2^{-l} < \Delta_t \leq 2^{-l+1}\right\}$$

$$\leq O(1) + T\Gamma + \mathbb{E}\sum_{t=1}^{T}\sum_{l=1}^{\lceil Q\rceil}2^{-l+1} \cdot \mathbb{1}\left\{\|X_t\|_{V_{t-1}^{-1}}^2 \geq \Omega\Big(\frac{2^{-2l}}{\beta_{t-1}\log(T\cdot 2^{2l})}\Big)\right\}$$

$$= O(1) + T\Gamma + \mathbb{E}\sum_{l=1}^{\lceil Q\rceil}2^{-l+1}\sum_{t=1}^{T}\mathbb{1}\left\{\|X_t\|_{V_{t-1}^{-1}}^2 \geq \Omega\Big(\frac{2^{-2l}}{\beta_{t-1}\log(T\cdot 2^{2l})}\Big)\right\}$$

$$\leq O(1) + T\Gamma + \mathbb{E}\sum_{l=1}^{\lceil Q\rceil}2^{-l+1}O\Big(2^{2l}d\beta_T\log(T\cdot 2^{2l})\log\Big(1 + \frac{2L^2\cdot 2^{2l}\beta_T\log(T\cdot 2^{2l})}{\lambda}\Big)\Big)$$

$$< O(1) + T\Gamma + \mathbb{E}\sum_{l=1}^{\lceil Q\rceil}2^{l+1}\cdot O\Big(d\beta_T\log(\frac{T}{\Gamma^2})\log\Big(1 + \frac{L^2\beta_T\log(\frac{T}{\Gamma^2})}{\lambda\Gamma^2}\Big)\Big)$$

$$\leq O(1) + T\Gamma + O\Big(\frac{d\beta_T\log(\frac{T}{\Gamma^2})}{\Gamma}\log\Big(1 + \frac{L^2\beta_T\log(\frac{T}{\Gamma^2})}{\lambda\Gamma^2}\Big)\Big),$$

This proves Eqn. (12). Then with the choice of the parameters as in Eqn. (5),

$$\mathbb{E}\sum_{t=1}^{T}\Delta_t \cdot \mathbb{1}\left\{B_t, C_t, D_t\right\} \cdot \mathbb{1}\left\{\|X_t\|_{V_{t-1}^{-1}}^2 < \frac{\Delta_t^2}{\beta_{t-1}}\right\}$$

$$< O(1) + d\sqrt{T}\log^{\frac{3}{2}}T + O\Big(d\beta_T\log\Big(\frac{T^2}{d^2\log^3 T}\Big)\frac{\sqrt{T}}{d\log^{\frac{3}{2}}T}\log\Big(1 + \frac{L^2\beta_T T}{\lambda d^2\log^3 T}\cdot\log\Big(\frac{T^2}{d^2\log^3 T}\Big)\Big)\Big)$$

$$\leq O\Big(d\sqrt{T}\log^{\frac{3}{2}}T\Big).$$

Hence, we can upper bound $F_1$ as

$$F_1 = \mathbb{E}\sum_{t=1}^{T}\Delta_t \cdot \mathbb{1}\left\{B_t, C_t, D_t\right\} \cdot \mathbb{1}\left\{\|X_t\|_{V_{t-1}^{-1}}^2 \geq \frac{\Delta_t^2}{\beta_{t-1}}\right\} + \mathbb{E}\sum_{t=1}^{T}\Delta_t \cdot \mathbb{1}\left\{B_t, C_t, D_t\right\} \cdot \mathbb{1}\left\{\|X_t\|_{V_{t-1}^{-1}}^2 < \frac{\Delta_t^2}{\beta_{t-1}}\right\}$$

$$\leq O\Big(d\sqrt{T}\log^{\frac{3}{2}}T\Big) + O\Big(d\sqrt{T}\log^{\frac{3}{2}}T\Big)$$

$$\leq O\Big(d\sqrt{T}\log^{\frac{3}{2}}T\Big),$$

which concludes the proof. $\qquad\square$

### C.3   Proof of Lemma 3

*Proof.* Since $C_t$ and $\overline{D}_t$ together imply that $\langle\theta^*, x_t^*\rangle - \delta < \varepsilon + \langle\hat{\theta}_{t-1}, X_t\rangle$, then using the choices of $\delta$ and $\varepsilon$, we have $\langle\hat{\theta}_{t-1} - \theta^*, X_t\rangle > \frac{\Delta_t}{\sqrt{\log T}}$. Substituting this into the event $B_t$ and using the Cauchy–Schwarz inequality, we have

$$\|X_t\|_{V_{t-1}^{-1}}^2 \geq \frac{\Delta_t^2}{\beta_{t-1}(\gamma)\log T}.$$

Again applying the "peeling device" on $\Delta_t$ and Corollary 1, we can upper bound $F_2$ as follows:

$$F_2 \leq \mathbb{E}\sum_{t=1}^{T}\Delta_t \cdot \mathbb{1}\left\{\|X_t\|_{V_{t-1}^{-1}}^2 \geq \frac{\Delta_t^2}{\beta_{t-1}\log T}\right\}$$

$$\leq T\Gamma + \mathbb{E} \sum_{t=1}^{T} \sum_{l=1}^{\lceil Q \rceil} \Delta_t \cdot \mathbb{1}\left\{ \|X_t\|_{V_{t-1}^{-1}}^2 \geq \frac{\Delta_t^2}{\beta_{t-1} \log T} \right\} \cdot \mathbb{1}\left\{ 2^{-l} < \Delta_t \leq 2^{-l+1} \right\}$$

$$\leq T\Gamma + \mathbb{E} \sum_{t=1}^{T} \sum_{l=1}^{\lceil Q \rceil} 2^{-l+1} \cdot \mathbb{1}\left\{ \|X_t\|_{V_{t-1}^{-1}}^2 \geq \frac{2^{-2l}}{\beta_T \log T} \right\}$$

$$\leq T\Gamma + \mathbb{E} \sum_{l=1}^{\lceil Q \rceil} 2^{-l+1} \cdot 2^{2l} \cdot 6d\beta_T (\log T) \log\left( 1 + \frac{2^{2l+1} \cdot L^2 \beta_T \log T}{\lambda} \right)$$

$$\leq T\Gamma + \mathbb{E} \sum_{l=1}^{\lceil Q \rceil} 2^{l} \cdot 12d\beta_T (\log T) \log\left( 1 + \frac{2^{2\lceil Q \rceil+1} \cdot L^2 \beta_T \log T}{\lambda} \right)$$

$$= T\Gamma + (2^{\lceil Q \rceil} - 1) \cdot 24d\beta_T (\log T) \log\left( 1 + \frac{2^{2\lceil Q \rceil+1} \cdot L^2 \beta_T \log T}{\lambda} \right)$$

$$< T\Gamma + \frac{48d\beta_T \log T}{\Gamma} \log\left( 1 + \frac{8L^2 \beta_T \log T}{\lambda \Gamma^2} \right)$$

$$= T\Gamma + O\left( \frac{d\beta_T \log T}{\Gamma} \log\left( 1 + \frac{L^2 \beta_T \log T}{\lambda \Gamma^2} \right) \right)$$

This proves Eqn. (13). Hence with the choice of the parameter $\Gamma$ as in Eqn. (5),

$$F_2 \leq d\sqrt{T} \log^{\frac{3}{2}} T + O\left( d\sqrt{T} \log^{\frac{3}{2}} T \right)$$

$$\leq O\left( d\sqrt{T} \log^{\frac{3}{2}} T \right) .$$

$\square$

## C.4   Proof of Lemma 4

*Proof.* For $F_3$, this is the case when the best arm at time $t$ does not perform sufficiently well so that the empirically largest reward at time $t$ is far from the highest expected reward. One observes that minimizing $F_3$ results in a tradeoff with respect to $F_1$. On the event $\overline{C}_t$, we can apply the "peeling device" on $\langle \theta^*, x_t^* \rangle - \langle \hat{\theta}_{t-1}, x_t^* \rangle$ such that $\frac{q+1}{2}\delta \leq \langle \theta^*, x_t^* \rangle - \langle \hat{\theta}_{t-1}, x_t^* \rangle < \frac{q+2}{2}\delta$ where $q \in \mathbb{N}$. Then using the fact that $I_{t,A_t} \leq I_{t,a_t^*}$, we have

$$\log \frac{1}{\beta_{t-1}\|X_t\|_{V_{t-1}^{-1}}^2} < \frac{q^2\delta^2}{4\beta_{t-1}\|x_t^*\|_{V_{t-1}^{-1}}^2} + \log \frac{1}{\beta_{t-1}\|x_t^*\|_{V_{t-1}^{-1}}^2} . \tag{17}$$

On the other hand, using the event $B_t$ and the Cauchy–Schwarz inequality, it holds that

$$\|x_t^*\|_{V_{t-1}^{-1}} \geq \frac{(q+1)\delta}{2\sqrt{\beta_{t-1}}} . \tag{18}$$

If $\|X_t\|_{V_{t-1}^{-1}}^2 \geq \frac{\Delta_t^2}{\beta_{t-1}}$, the regret in this case is bounded by $O(d\sqrt{T \log T})$ (similar to the procedure to get from Eqn. (15) to Eqn. (16)). Otherwise $\log \frac{1}{\beta_{t-1}\|X_t\|_{V_{t-1}^{-1}}^2} > \log \frac{1}{\Delta_t^2} \geq 0$, then combining Eqn. (17) and Eqn. (18) implies that

$$\|X_t\|_{V_{t-1}^{-1}}^2 \geq \frac{(q+1)^2\delta^2}{4\beta_{t-1}} \exp\left( -\frac{q^2}{(q+1)^2} \right) .$$

Notice here with $\sqrt{\lambda}S \geq 1$, $\|X_t\|_{V_{t-1}^{-1}}^2 < \frac{\Delta_t^2}{\beta_{t-1}} \leq \frac{1}{\beta_{t-1}} \leq 1$, it holds that for all $q \in \mathbb{N}$,

$$\frac{(q+1)^2\delta^2}{4\beta_{t-1}} \exp\left( -\frac{q^2}{(q+1)^2} \right) < 1 . \tag{19}$$

Using Corollary 1, one can show that:

$$\sum_{t=1}^{T} \Delta_t \cdot \mathbb{1}\left\{B_t, \overline{C}_t\right\} \cdot \mathbb{1}\left\{\|X_t\|_{V_{t-1}^{-1}}^2 < \frac{\Delta_t^2}{\beta_{t-1}}\right\}$$

$$\leq T\Gamma + \sum_{t=1}^{T} \sum_{l=1}^{\lceil Q \rceil} \Delta_t \cdot \mathbb{1}\left\{B_t, \overline{C}_t\right\} \cdot \mathbb{1}\left\{\|X_t\|_{V_{t-1}^{-1}}^2 < \frac{\Delta_t^2}{\beta_{t-1}}\right\} \cdot \mathbb{1}\left\{2^{-l} < \Delta_t \leq 2^{-l+1}\right\}$$

$$\leq T\Gamma + \sum_{t=1}^{T} \sum_{l=1}^{\lceil Q \rceil} \sum_{q=1}^{\infty} \Delta_t \cdot \mathbb{1}\left\{B_t\right\} \cdot \mathbb{1}\left\{\frac{q+1}{2}\delta \leq \langle \theta^*, x_t^* \rangle - \langle \hat{\theta}_{t-1}, x_t^* \rangle < \frac{q+2}{2}\delta\right\} \cdot \mathbb{1}\left\{\|X_t\|_{V_{t-1}^{-1}}^2 < \frac{\Delta_t^2}{\beta_{t-1}}\right\}$$
$$\cdot \mathbb{1}\left\{2^{-l} < \Delta_t \leq 2^{-l+1}\right\}$$

$$\leq T\Gamma + \sum_{t=1}^{T} \sum_{l=1}^{\lceil Q \rceil} \sum_{q=1}^{\infty} \Delta_t \cdot \mathbb{1}\left\{1 \geq \|X_t\|_{V_{t-1}^{-1}}^2 \geq \frac{(q+1)^2\delta^2}{4\beta_{t-1}} \exp\left(-\frac{q^2}{(q+1)^2}\right)\right\} \cdot \mathbb{1}\left\{2^{-l} < \Delta_t \leq 2^{-l+1}\right\}$$

$$= T\Gamma + \sum_{t=1}^{T} \sum_{l=1}^{\lceil Q \rceil} \sum_{q=1}^{\infty} \Delta_t \cdot \mathbb{1}\left\{1 \geq \|X_t\|_{V_{t-1}^{-1}}^2 \geq \frac{(q+1)^2\Delta_t^2}{4\beta_{t-1}\log T} \exp\left(-\frac{q^2}{(q+1)^2}\right)\right\} \cdot \mathbb{1}\left\{2^{-l} < \Delta_t \leq 2^{-l+1}\right\}$$

$$\leq T\Gamma + \sum_{t=1}^{T} \sum_{l=1}^{\lceil Q \rceil} \sum_{q=1}^{\infty} 2^{-l+1} \cdot \mathbb{1}\left\{1 \geq \|X_t\|_{V_{t-1}^{-1}}^2 > \frac{(q+1)^2 \cdot 2^{-2l}}{4\beta_T \log T} \exp\left(-\frac{q^2}{(q+1)^2}\right)\right\}$$

$$\leq T\Gamma + \sum_{l=1}^{\lceil Q \rceil} \sum_{q=1}^{\infty} 2^{-l+1} \cdot 2^{2l} \cdot 24 d\beta_T (\log T) \cdot \frac{\exp\left(\frac{q^2}{(q+1)^2}\right)}{(q+1)^2} \cdot \log\left(1 + \frac{2^{2l} \cdot 8L^2\beta_T \log T}{\lambda} \cdot \frac{\exp\left(\frac{q^2}{(q+1)^2}\right)}{(q+1)^2}\right)$$

$$< T\Gamma + \sum_{l=1}^{\lceil Q \rceil} \sum_{q=1}^{\infty} 2^{l+1} \cdot 24 d\beta_T (\log T) \cdot \frac{\exp\left(\frac{q^2}{(q+1)^2}\right)}{(q+1)^2} \cdot \log\left(1 + \frac{2^{2l+1} \cdot L^2\beta_T \log T}{\lambda}\right)$$

$$= T\Gamma + \sum_{l=1}^{\lceil Q \rceil} 2^{l+1} \cdot 24 d\beta_T (\log T) \cdot \log\left(1 + \frac{2^{2l+1} \cdot L^2\beta_T \log T}{\lambda}\right) \sum_{q=1}^{\infty} \frac{\exp\left(\frac{q^2}{(q+1)^2}\right)}{(q+1)^2}$$

$$\leq T\Gamma + \sum_{l=1}^{\lceil Q \rceil} 2^{l+1} \cdot 24 d\beta_T (\log T) \cdot \log\left(1 + \frac{2^{2l+1} \cdot L^2\beta_T \log T}{\lambda}\right) \cdot (1.09)$$

$$\leq T\Gamma + \sum_{l=1}^{\lceil Q \rceil} 2^{l+1} \cdot 27 d\beta_T (\log T) \cdot \log\left(1 + \frac{2^{2l+1} \cdot L^2\beta_T \log T}{\lambda}\right)$$

$$\leq T\Gamma + \sum_{l=1}^{\lceil Q \rceil} 2^{l+1} \cdot 27 d\beta_T (\log T) \cdot \log\left(1 + \frac{2^{2\lceil Q \rceil+1} \cdot L^2\beta_T \log T}{\lambda}\right)$$

$$< T\Gamma + \sum_{l=1}^{\lceil Q \rceil} \frac{216 d\beta_T \log T}{\Gamma} \cdot \log\left(1 + \frac{8L^2\beta_T \log T}{\lambda\Gamma^2}\right)$$

$$= T\Gamma + O\left(\frac{d\beta_T \log T}{\Gamma} \log\left(1 + \frac{L^2\beta_T \log T}{\lambda\Gamma^2}\right)\right). \tag{20}$$

Hence

$$F_3 = \sum_{t=1}^{T} \Delta_t \cdot \mathbb{1}\left\{B_t, \overline{C}_t\right\} \cdot \mathbb{1}\left\{\|X_t\|_{V_{t-1}^{-1}}^2 < \frac{\Delta_t^2}{\beta_{t-1}}\right\} + \sum_{t=1}^{T} \Delta_t \cdot \mathbb{1}\left\{B_t, \overline{C}_t\right\} \cdot \mathbb{1}\left\{\|X_t\|_{V_{t-1}^{-1}}^2 \geq \frac{\Delta_t^2}{\beta_{t-1}}\right\}$$

$$< O\left(\frac{d\beta_T}{\Gamma} \log\left(1 + \frac{L^2\beta_T}{\lambda\Gamma^2}\right)\right) + 2T\Gamma + O\left(\frac{d\beta_T \log T}{\Gamma} \log\left(1 + \frac{L^2\beta_T \log T}{\lambda\Gamma^2}\right)\right)$$

$$\le 2T\Gamma + O\left(\frac{d\beta_T \log(T)}{\Gamma} \log\left(1 + \frac{L^2\beta_T \log(T)}{\lambda\Gamma^2}\right)\right).$$

This proves Eqn. (14). With the choice of $\Gamma$ as in Eqn. (5),

$$F_3 \le 2d\sqrt{T}\log^{\frac{3}{2}} T + O\left(\frac{d\sqrt{T}\beta_T \log T}{d\log^{\frac{3}{2}} T} \log\left(1 + \frac{TL^2\beta_T \log T}{\lambda d^2 \log^3 T}\right)\right)$$

$$< 2d\sqrt{T}\log^{\frac{3}{2}} T + O\left(d\sqrt{T}\log^{\frac{3}{2}} T\right)$$

$$= O\left(d\sqrt{T}\log^{\frac{3}{2}} T\right).$$

$\square$

### C.5 Proof of Lemma 5

*Proof.* For $F_4$, the proof is straightforward by using Lemma 1 with the choice of $\gamma$. Indeed, one has

$$F_4 = \mathbb{E}\sum_{t=1}^{T} \Delta_t \cdot \mathbb{1}\left\{\overline{B}_t\right\} \le T\Gamma + \mathbb{E}\sum_{t=1}^{T}\sum_{l=1}^{\lceil Q \rceil} \Delta_t \cdot \mathbb{1}\left\{2^{-l} < \Delta_t \le 2^{-l+1}\right\} \mathbb{1}\left\{\overline{B}_t\right\}$$

$$\le T\Gamma + \mathbb{E}\sum_{t=1}^{T}\sum_{l=1}^{\lceil Q \rceil} 2^{-l+1}\mathbb{1}\left\{\overline{B}_t\right\} \le T\Gamma + \sum_{t=1}^{T}\sum_{l=1}^{\lceil Q \rceil} 2^{-l+1}\mathbb{P}(\overline{B}_t) \le T\Gamma + \sum_{t=1}^{T}\sum_{l=1}^{\lceil Q \rceil} 2^{-l+1}\gamma$$

$$= T\Gamma + \sum_{t=1}^{T}\frac{1}{t^2}\sum_{l=1}^{\lceil Q \rceil} 2^{-l+1} = T\Gamma + \sum_{t=1}^{T}\frac{2-\Gamma}{t^2} < T\Gamma + \frac{\pi^2}{3} = T\Gamma + O(1).$$

With the choice of $\Gamma$ as in Eqn. (5),

$$F_4 < d\sqrt{T}\log^{\frac{3}{2}} T + O(1)$$

$$\le O\left(d\sqrt{T}\log^{\frac{3}{2}} T\right).$$

$\square$

### C.6 Proof of Theorem 1

*Proof.* Combining Lemmas 2, 3, 4 and 5,

$$R_T = F_1 + F_2 + F_3 + F_4$$

$$\le O\left(d\sqrt{T}\log^{\frac{3}{2}} T\right) + O\left(d\sqrt{T}\log^{\frac{3}{2}} T\right) + O\left(d\sqrt{T}\log^{\frac{3}{2}} T\right) + O\left(d\sqrt{T}\log^{\frac{3}{2}} T\right)$$

$$= O\left(d\sqrt{T}\log^{\frac{3}{2}} T\right).$$

$\square$

## D Proof of the regret bound for LinIMED-2 (Proof of Theorem 2)

We choose $\gamma$ and $\Gamma$ as follows:

$$\gamma = \frac{1}{t^2} \qquad \Gamma = \frac{\sqrt{d\beta_T}\log T}{\sqrt{T}}. \tag{21}$$

### D.1  Statement of Lemmas for LinIMED-2

We first state the following lemmas which respectively show the upper bound of $F_1$ to $F_4$:

**Lemma 6.** *Under Assumption 1, and the assumption that $\sqrt{\lambda}S \geq 1$, for the free parameter $0 < \Gamma < 1$, the term $F_1$ for LinIMED-3 satisfies:*

$$F_1 \leq T\Gamma + O\left(\frac{d\beta_T \log T}{\Gamma}\right) \log\left(1 + \frac{L^2\beta_T \log T}{\lambda\Gamma^2}\right) .$$

**Lemma 7.** *Under Assumption 1, and the assumption that $\sqrt{\lambda}S \geq 1$, for the free parameter $0 < \Gamma < 1$, the term $F_2$ for LinIMED-3 satisfies:*

$$F_2 \leq T\Gamma + O\left(\frac{d\beta_T \log T}{\Gamma}\right) \log\left(1 + \frac{L^2\beta_T \log T}{\lambda\Gamma^2}\right) .$$

**Lemma 8.** *Under Assumption 1, and the assumption that $\sqrt{\lambda}S \geq 1$, for the free parameter $0 < \Gamma < 1$, the term $F_3$ for LinIMED-3 satisfies:*

$$F_3 \leq 5T\Gamma + O\left(\frac{d\beta_T \log T}{\Gamma} \log\left(1 + \frac{L^2\beta_T \log T}{\lambda\Gamma^2}\right)\right) + O\left(\sqrt{T \log T} \log\left(\frac{L^2\beta_T \log T}{\lambda\Gamma^2}\right)\right) .$$

**Lemma 9.** *Under Assumption 1, with the choice of $\gamma = \frac{1}{t^2}$ as in Eqn. (21), for the free parameter $0 < \Gamma < 1$, the term $F_4$ for LinIMED-3 satisfies:*

$$F_4 \leq T\Gamma + O(1) .$$

### D.2  Proof of Lemma 6

*Proof.* We first partition the analysis into the cases $\hat{A}_t \neq A_t$ and $\hat{A}_t = A_t$ as follows:

$$\begin{aligned}
F_1 &= \mathbb{E}\sum_{t=1}^{T} \Delta_t \cdot \mathbb{1}\{B_t, C_t, D_t\} \\
&= \mathbb{E}\sum_{t=1}^{T} \Delta_t \cdot \mathbb{1}\{B_t, C_t, D_t\} \cdot \mathbb{1}\left\{\hat{A}_t \neq A_t\right\} + \mathbb{E}\sum_{t=1}^{T} \Delta_t \cdot \mathbb{1}\{B_t, C_t, D_t\} \cdot \mathbb{1}\left\{\hat{A}_t = A_t\right\}
\end{aligned}$$

**Case 1:** If $\hat{A}_t \neq A_t$, this means that the index of $A_t$ is $I_{t,A_t} = \frac{\hat{\Delta}_{t,A_t}^2}{\beta_{t-1}\|X_t\|_{V_{t-1}^{-1}}^2} + \log\frac{1}{\beta_{t-1}\|X_t\|_{V_{t-1}^{-1}}^2}$. Using the fact that $I_{t,A_t} \leq I_{t,\hat{A}_t}$ we have:

$$\begin{aligned}
I_{t,A_t} &= \frac{\hat{\Delta}_{t,A_t}^2}{\beta_{t-1}\|X_t\|_{V_{t-1}^{-1}}^2} + \log\frac{1}{\beta_{t-1}\|X_t\|_{V_{t-1}^{-1}}^2} \\
&\leq \log T \wedge \log\frac{1}{\beta_{t-1}\|\hat{X}_t\|_{V_{t-1}^{-1}}^2} \\
&\leq \log T.
\end{aligned}$$

Therefore

$$\frac{\hat{\Delta}_{t,A_t}^2}{\beta_{t-1}\|X_t\|_{V_{t-1}^{-1}}^2} + \log\frac{1}{\beta_{t-1}\|X_t\|_{V_{t-1}^{-1}}^2} \leq \log T . \tag{22}$$

If $\|X_t\|^2_{V^{-1}_{t-1}} \geq \frac{\Delta^2_t}{\beta_{t-1}}$, using the same procedure to get from Eqn. (15) to Eqn. (16), one has:

$$\mathbb{E}\sum_{t=1}^{T}\Delta_t \cdot \mathbb{1}\{B_t, C_t, D_t\} \cdot \mathbb{1}\left\{\hat{A}_t \neq A_t\right\} \cdot \mathbb{1}\left\{\|X_t\|^2_{V^{-1}_{t-1}} \geq \frac{\Delta^2_t}{\beta_{t-1}}\right\}$$

$$\leq \mathbb{E}\sum_{t=1}^{T}\Delta_t \cdot \mathbb{1}\left\{\|X_t\|^2_{V^{-1}_{t-1}} \geq \frac{\Delta^2_t}{\beta_{t-1}}\right\}$$

$$< T\Gamma + \frac{48d\beta_T}{\Gamma}\log\left(1 + \frac{8L^2\beta_T}{\lambda\Gamma^2}\right)$$

$$= T\Gamma + O\left(\frac{d\beta_T}{\Gamma}\log\left(1 + \frac{L^2\beta_T}{\lambda\Gamma^2}\right)\right).$$

Else if $\|X_t\|^2_{V^{-1}_{t-1}} < \frac{\Delta^2_t}{\beta_{t-1}}$, this implies that $\log\frac{1}{\beta_{t-1}\|X_t\|^2_{V^{-1}_{t-1}}} > \log\frac{1}{\Delta^2_t} \geq 0$. Then substituting the event $D_t := \{\hat{\Delta}_{t,A_t} \geq \varepsilon\}$ into Eqn. (22), we obtain

$$\frac{\varepsilon^2}{\beta_{t-1}\|X_t\|^2_{V^{-1}_{t-1}}} \leq \log T .$$

With $\sqrt{\lambda}S \geq 1$ we have $\beta_{t-1} \geq 1$ , then one has

$$\|X_t\|^2_{V^{-1}_{t-1}} \geq \frac{\varepsilon^2}{\beta_{t-1}\log T}.$$

Hence

$$\mathbb{E}\sum_{t=1}^{T}\Delta_t \cdot \mathbb{1}\left\{B_t, C_t, D_t, \hat{A}_t \neq A_t, \|X_t\|^2_{V^{-1}_{t-1}} < \frac{\Delta^2_t}{\beta_{t-1}}\right\}$$

$$\leq \mathbb{E}\sum_{t=1}^{T}\Delta_t \cdot \mathbb{1}\left\{\|X_t\|^2_{V^{-1}_{t-1}} \geq \frac{\varepsilon^2}{\beta_{t-1}\log T}\right\}.$$

With the choice of $\varepsilon = (1 - \frac{2}{\sqrt{\log T}})\Delta_t$, when $T \geq 149 > \exp(5)$, $\varepsilon > \frac{\Delta_t}{10}$, then performing the "peeling device" on $\Delta_t$ yields

$$\mathbb{E}\sum_{t=1}^{T}\Delta_t \cdot \mathbb{1}\left\{\|X_t\|^2_{V^{-1}_{t-1}} \geq \frac{\varepsilon^2}{\beta_{t-1}\log T}\right\} \cdot \mathbb{1}\{\Delta_t \geq \Gamma\}$$

$$\leq 149 + \mathbb{E}\sum_{t=1}^{T}\sum_{l=1}^{\lceil Q\rceil}\Delta_t \cdot \mathbb{1}\left\{2^{-l} < \Delta_t \leq 2^{-l+1}, \|X_t\|^2_{V^{-1}_{t-1}} \geq \frac{\varepsilon^2}{\beta_{t-1}\log T}\right\}$$

$$\leq O(1) + \mathbb{E}\sum_{l=1}^{\lceil Q\rceil}2^{-l+1}\sum_{t=1}^{T}\mathbb{1}\left\{\|X_t\|^2_{V^{-1}_{t-1}} \geq \frac{\varepsilon^2}{\beta_{t-1}\log T}\right\}$$

$$\leq O(1) + \mathbb{E}\sum_{l=1}^{\lceil Q\rceil}2^{-l+1}\sum_{t=1}^{T}\mathbb{1}\left\{\|X_t\|^2_{V^{-1}_{t-1}} \geq \frac{2^{-2l}}{100\beta_T\log T}\right\}$$

$$\leq O(1) + \mathbb{E}\sum_{l=1}^{\lceil Q\rceil}2^{-l+1} \cdot 2^{2l} \cdot 600d\beta_T(\log T)\log\left(1 + \frac{2^{2l} \cdot 200L^2\beta_T\log T}{\lambda}\right)$$

$$\leq O(1) + \mathbb{E} \sum_{l=1}^{\lceil Q \rceil} 2^{l+1} \cdot 600 d\beta_T (\log T) \log \left( 1 + \frac{2^{2\lceil Q \rceil} \cdot 200 L^2 \beta_T \log T}{\lambda} \right)$$

$$< O(1) + \frac{4800 d\beta_T \log T}{\Gamma} \log \left( 1 + \frac{800 L^2 \beta_T \log T}{\lambda \Gamma^2} \right).$$

Considering the event $\{\Delta_t < \Gamma\}$, we can upper bound the corresponding expectation as follows

$$\mathbb{E} \sum_{t=1}^{T} \Delta_t \cdot \mathbb{1} \left\{ \|X_t\|_{V_{t-1}^{-1}}^2 \geq \frac{\varepsilon^2}{\beta_{t-1} \log T} \right\} \cdot \mathbb{1} \{\Delta_t < \Gamma\} \leq \mathbb{E} \sum_{t=1}^{T} \Delta_t \cdot \mathbb{1} \{\Delta_t < \Gamma\} < T\Gamma.$$

Then

$$\mathbb{E} \sum_{t=1}^{T} \Delta_t \cdot \mathbb{1} \left\{ B_t, C_t, D_t, \hat{A}_t \neq A_t, \|X_t\|_{V_{t-1}^{-1}}^2 < \frac{\Delta_t^2}{\beta_{t-1}} \right\}$$

$$\leq \mathbb{E} \sum_{t=1}^{T} \Delta_t \cdot \mathbb{1} \left\{ \|X_t\|_{V_{t-1}^{-1}}^2 \geq \frac{\varepsilon^2}{\beta_{t-1} \log T} \right\}$$

$$= \mathbb{E} \sum_{t=1}^{T} \Delta_t \cdot \mathbb{1} \left\{ \|X_t\|_{V_{t-1}^{-1}}^2 \geq \frac{\varepsilon^2}{\beta_{t-1} \log T} \right\} \cdot \mathbb{1} \{\Delta_t \geq \Gamma\}$$

$$+ \mathbb{E} \sum_{t=1}^{T} \Delta_t \cdot \mathbb{1} \left\{ \|X_t\|_{V_{t-1}^{-1}}^2 \geq \frac{\varepsilon^2}{\beta_{t-1} \log T} \right\} \cdot \mathbb{1} \{\Delta_t < \Gamma\}$$

$$\leq O(1) + T\Gamma + \frac{4800 d\beta_T \log T}{\Gamma} \log \left( 1 + \frac{800 L^2 \beta_T \log T}{\lambda \Gamma^2} \right).$$

Hence

$$\mathbb{E} \sum_{t=1}^{T} \Delta_t \cdot \mathbb{1} \left\{ B_t, C_t, D_t, \hat{A}_t \neq A_t \right\}$$

$$= \mathbb{E} \sum_{t=1}^{T} \Delta_t \cdot \mathbb{1} \left\{ B_t, C_t, D_t, \hat{A}_t \neq A_t, \|X_t\|_{V_{t-1}^{-1}}^2 \geq \frac{\Delta_t^2}{\beta_{t-1}} \right\}$$

$$+ \mathbb{E} \sum_{t=1}^{T} \Delta_t \cdot \mathbb{1} \left\{ B_t, C_t, D_t, \hat{A}_t \neq A_t, \|X_t\|_{V_{t-1}^{-1}}^2 < \frac{\Delta_t^2}{\beta_{t-1}} \right\}$$

$$\leq T\Gamma + O\left( \frac{d\beta_T}{\Gamma} \log \left( 1 + \frac{L^2 \beta_T}{\lambda \Gamma^2} \right) \right) + O(1) + T\Gamma + \frac{4800 d\beta_T \log T}{\Gamma} \log \left( 1 + \frac{800 L^2 \beta_T \log T}{\lambda \Gamma^2} \right)$$

$$\leq T\Gamma + O\left( \frac{d\beta_T \log T}{\Gamma} \log \left( 1 + \frac{L^2 \beta_T \log T}{\lambda \Gamma^2} \right) \right).$$

**Case 2:** If $\hat{A}_t = A_t$, then from the event $C_t$ and the choice $\delta = \frac{\Delta_t}{\sqrt{\log T}}$ we have

$$\langle \hat{\theta}_{t-1} - \theta^*, X_t \rangle > \left( 1 - \frac{1}{\sqrt{\log T}} \right) \Delta_t.$$

Furthermore, using the definition of the event $B_t$, that implies that

$$\|X_t\|_{V_{t-1}^{-1}}^2 > \frac{(1 - \frac{1}{\sqrt{\log T}})^2 \Delta_t^2}{\beta_{t-1}}.$$

When $T > 8 > \exp(2)$, $(1 - \frac{1}{\sqrt{\log T}})^2 > \frac{1}{16}$, then similarily, we can bound this term by $O(\frac{d\beta_T}{\Gamma}) \log(1 + \frac{L^2 \beta_T}{\lambda \Gamma^2})$

Summarizing the two cases,

$$F_1 \le O(1) + T\Gamma + O\left(\frac{d\beta_T \log T}{\Gamma}\right) \log\left(1 + \frac{L^2 \beta_T \log T}{\lambda \Gamma^2}\right)$$
$$\le T\Gamma + O\left(\frac{d\beta_T \log T}{\Gamma}\right) \log\left(1 + \frac{L^2 \beta_T \log T}{\lambda \Gamma^2}\right).$$

$\square$

### D.3 Proof of Lemma 7

*Proof.* Recall that

$$F_2 = \mathbb{E} \sum_{t=1}^{T} \Delta_t \cdot \mathbb{1}\left\{B_t, C_t, \overline{D}_t\right\}.$$

From $C_t$ and $\overline{D}_t$, we derive that:

$$\langle \theta^*, a_t^* \rangle - \delta < \varepsilon + \langle \hat{\theta}_{t-1}, X_t \rangle.$$

With the choice $\delta = \frac{\Delta_t}{\sqrt{\log T}}, \varepsilon = (1 - \frac{2}{\sqrt{\log T}})\Delta_t$, we have

$$\langle \hat{\theta}_{t-1} - \theta^*, X_t \rangle > \frac{\Delta_t}{\sqrt{\log T}}. \tag{23}$$

Then using the definition of the event $B_t$ in Eqn. (23) yields

$$\|X_t\|_{V_{t-1}^{-1}}^2 \ge \frac{\Delta_t^2}{\beta_{t-1} \log T}.$$

Using a similar procedure as in that from Eqn. (15) to Eqn. (16), we can upper bound $F_2$ by

$$F_2 \le T\Gamma + O\left(\frac{d\beta_T \log T}{\Gamma}\right) \log\left(1 + \frac{L^2 \beta_T \log T}{\lambda \Gamma^2}\right).$$

$\square$

### D.4 Proof of Lemma 8

*Proof.* From the event $\overline{C}_t$, which is $\max_{b \in \mathcal{A}_t}\langle \hat{\theta}_{t-1}, b \rangle \le \langle \theta^*, x_t^* \rangle - \delta$, the index of the best arm at time $t$ can be upper bounded as:

$$I_{t,a_t^*} \le \frac{(\langle \theta^*, x_t^* \rangle - \delta - \langle \hat{\theta}_{t-1}, x_t^* \rangle)^2}{\beta_{t-1}\|x_t^*\|_{V_{t-1}^{-1}}^2} + \log \frac{1}{\beta_{t-1}\|x_t^*\|_{V_{t-1}^{-1}}^2}.$$

**Case 1:** If $\hat{A}_t \ne A_t$, then we have

$$I_{t,a_t^*} \ge I_{t,A_t} \ge \log \frac{1}{\beta_{t-1}\|X_t\|_{V_{t-1}^{-1}}^2}.$$

Suppose $\frac{q+1}{2}\delta \le \langle \theta^*, x_t^* \rangle - \langle \hat{\theta}_{t-1}, x_t^* \rangle < \frac{q+2}{2}\delta$ for $q \in \mathbb{N}$, then one has

$$\log \frac{1}{\beta_{t-1}\|X_t\|_{V_{t-1}^{-1}}^2} \le \frac{q^2 \delta^2}{4\beta_{t-1}\|x_t^*\|_{V_{t-1}^{-1}}^2} + \log \frac{1}{\beta_{t-1}\|x_t^*\|_{V_{t-1}^{-1}}^2}. \tag{24}$$

On the other hand, on the event $B_t$,

$$\|x_t^*\|_{V_{t-1}^{-1}} \geq \frac{(q+1)\delta}{2\sqrt{\beta_{t-1}}}. \tag{25}$$

If $\|X_t\|_{V_{t-1}^{-1}}^2 \geq \frac{\Delta_t^2}{\beta_{t-1}}$, using the same procedure from Eqn. (15) to Eqn. (16), one has:

$$\mathbb{E}\sum_{t=1}^T \Delta_t \cdot \mathbb{1}\left\{B_t, \overline{C}_t\right\} \cdot \mathbb{1}\left\{\hat{A}_t \neq A_t\right\} \cdot \mathbb{1}\left\{\|X_t\|_{V_{t-1}^{-1}}^2 \geq \frac{\Delta_t^2}{\beta_{t-1}}\right\}$$

$$\leq \mathbb{E}\sum_{t=1}^T \Delta_t \cdot \mathbb{1}\left\{\|X_t\|_{V_{t-1}^{-1}}^2 \geq \frac{\Delta_t^2}{\beta_{t-1}}\right\}$$

$$< T\Gamma + \frac{48d\beta_T}{\Gamma}\log\left(1 + \frac{8L^2\beta_T}{\lambda\Gamma^2}\right)$$

$$= T\Gamma + O\left(\frac{d\beta_T}{\Gamma}\log\left(1 + \frac{L^2\beta_T}{\lambda\Gamma^2}\right)\right).$$

Else if $\|X_t\|_{V_{t-1}^{-1}}^2 < \frac{\Delta_t^2}{\beta_{t-1}}$, this implies that $\log\frac{1}{\beta_{t-1}\|X_t\|_{V_{t-1}^{-1}}^2} > \log\frac{1}{\Delta_t^2} \geq 0$. Then combining Eqn. (24) and Eqn. (25) implies that

$$\|X_t\|_{V_{t-1}^{-1}}^2 \geq \frac{(q+1)^2\delta^2}{4\beta_{t-1}}\exp\left(-\frac{q^2}{(q+1)^2}\right).$$

Then using the same procedure to get from Eqn. (19) to Eqn. (20), we have

$$\sum_{t=1}^T \Delta_t \cdot \mathbb{1}\left\{B_t, \overline{C}_t\right\} \cdot \mathbb{1}\left\{\|X_t\|_{V_{t-1}^{-1}}^2 < \frac{\Delta_t^2}{\beta_{t-1}}, \hat{A}_t \neq A_t\right\}$$

$$< T\Gamma + O\left(\frac{d\beta_T \log T}{\Gamma}\log\left(1 + \frac{L^2\beta_T \log T}{\lambda\Gamma^2}\right)\right). \tag{26}$$

**Case 2:** $\hat{A}_t = A_t$. If $\|X_t\|_{V_{t-1}^{-1}}^2 \geq \frac{\Delta_t^2}{\beta_{t-1}}$, using the same procedure to get from Eqn. (15) to Eqn. (16), one has:

$$\mathbb{E}\sum_{t=1}^T \Delta_t \cdot \mathbb{1}\left\{B_t, \overline{C}_t\right\} \cdot \mathbb{1}\left\{\hat{A}_t = A_t\right\} \cdot \mathbb{1}\left\{\|X_t\|_{V_{t-1}^{-1}}^2 \geq \frac{\Delta_t^2}{\beta_{t-1}}\right\}$$

$$\leq \mathbb{E}\sum_{t=1}^T \Delta_t \cdot \mathbb{1}\left\{\|X_t\|_{V_{t-1}^{-1}}^2 \geq \frac{\Delta_t^2}{\beta_{t-1}}\right\}$$

$$< T\Gamma + \frac{48d\beta_T}{\Gamma}\log\left(1 + \frac{8L^2\beta_T}{\lambda\Gamma^2}\right)$$

$$= T\Gamma + O\left(\frac{d\beta_T}{\Gamma}\log\left(1 + \frac{L^2\beta_T}{\lambda\Gamma^2}\right)\right).$$

Else $\|X_t\|_{V_{t-1}^{-1}}^2 < \frac{\Delta_t^2}{\beta_{t-1}}$ implies that $\log\frac{1}{\beta_{t-1}\|X_t\|_{V_{t-1}^{-1}}^2} > \log\frac{1}{\Delta_t^2} \geq 0$.

If $\log \frac{1}{\beta_{t-1} \|X_t\|^2_{V_{t-1}^{-1}}} < \log T$, then using the same procedure to get from Eqn. (24) to Eqn. (26), we have

$$\sum_{t=1}^{T} \Delta_t \cdot \mathbb{1}\left\{B_t, \overline{C}_t\right\} \cdot \mathbb{1}\left\{\|X_t\|^2_{V_{t-1}^{-1}} < \frac{\Delta_t^2}{\beta_{t-1}}, \hat{A}_t = A_t, \log \frac{1}{\beta_{t-1}\|X_t\|^2_{V_{t-1}^{-1}}} < \log \frac{T}{\beta_{t-1}}\right\}$$
$$< T\Gamma + O\left(\frac{d\beta_T \log T}{\Gamma} \log\left(1 + \frac{L^2 \beta_T \log T}{\lambda \Gamma^2}\right)\right).$$

If $\log \frac{1}{\beta_{t-1} \|X_t\|^2_{V_{t-1}^{-1}}} \geq \log T$, this means now the index of $A_t$ is $I_{t,A_t} = \log T$, by performing the "peeling device" such that $\frac{q+1}{2}\delta \leq \langle \theta^*, x_t^* \rangle - \langle \hat{\theta}_{t-1}, x_t^* \rangle < \frac{q+2}{2}\delta$ for $q \in \mathbb{N}$, we have

$$\log T \leq \frac{q^2 \delta^2}{4\beta_{t-1}\|x_t^*\|^2_{V_{t-1}^{-1}}} + \log \frac{1}{\beta_{t-1}\|x_t^*\|^2_{V_{t-1}^{-1}}}. \tag{27}$$

On the other hand, using the definition of the event $B_t$,

$$\|x_t^*\|_{V_{t-1}^{-1}} \geq \frac{(q+1)\delta}{2\sqrt{\beta_{t-1}}}. \tag{28}$$

Combining Eqn. (27) and (28), we have

$$\delta \leq \frac{2\exp(\frac{q^2}{2(q+1)^2})}{(q+1)\sqrt{T}}.$$

Then with $\delta = \frac{\Delta_t}{\sqrt{\log T}}$, this implies that

$$\Delta_t \leq \frac{2\sqrt{\log T}\exp(\frac{q^2}{2(q+1)^2})}{(q+1)\sqrt{T}}.$$

On the other hand, from $\frac{q+1}{2}\delta \leq \sqrt{\beta_{t-1}}\|x_t^*\|_{V_{t-1}^{-1}} \leq \sqrt{\beta_{t-1}} \cdot \frac{L}{\sqrt{\lambda}}$, we have $q+1 \leq \frac{2L\sqrt{\beta_{t-1}\log T}}{\sqrt{\lambda}\Delta_t}$. Hence,

$$\sum_{t=1}^{T} \Delta_t \cdot \mathbb{1}\left\{B_t, \overline{C}_t\right\} \cdot \mathbb{1}\left\{\|X_t\|^2_{V_{t-1}^{-1}} < \frac{\Delta_t^2}{\beta_{t-1}}, \hat{A}_t = A_t, \log \frac{1}{\beta_{t-1}\|X_t\|^2_{V_{t-1}^{-1}}} \geq \log T, \Delta_t \geq \Gamma\right\}$$

$$\leq \mathbb{E} \sum_{q=1}^{\lfloor \frac{2L\sqrt{\beta_T \log T}}{\sqrt{\lambda}\Gamma} - 1\rfloor} \sum_{t=1}^{T} \Delta_t \cdot \mathbb{1}\left\{\Delta_t \leq \frac{2\sqrt{\log T}\exp(\frac{q^2}{2(q+1)^2})}{(q+1)\sqrt{T}}\right\}$$

$$\leq \mathbb{E} \sum_{q=1}^{\lfloor \frac{2L\sqrt{\beta_T \log T}}{\sqrt{\lambda}\Gamma} - 1\rfloor} \sum_{t=1}^{T} \frac{2\sqrt{\log T}\exp(\frac{q^2}{2(q+1)^2})}{(q+1)\sqrt{T}}$$

$$= \mathbb{E} \sum_{q=1}^{\lfloor \frac{2L\sqrt{\beta_T \log T}}{\sqrt{\lambda}\Gamma} - 1\rfloor} \frac{2\sqrt{T\log T}\exp(\frac{q^2}{2(q+1)^2})}{q+1}$$

$$< \mathbb{E} \sum_{q=1}^{\lfloor \frac{2L\sqrt{\beta_T \log T}}{\sqrt{\lambda}\Gamma} - 1\rfloor} \frac{2\sqrt{e}\sqrt{T\log T}}{q+1}$$

$$< 2\sqrt{e}\sqrt{T \log T} \log \left( \frac{2L\sqrt{\log T}}{\sqrt{\lambda}\Gamma} - 1 \right)$$

$$\leq O\left( \sqrt{T \log T} \log \left( \frac{L^2 \beta_T \log T}{\lambda \Gamma^2} \right) \right) .$$

Summarizing the two cases ($\hat{A}_t \neq A_t$ and $\hat{A}_t = A_t$), we see that $F_3$ is upper bounded by:

$$F_3 < T\Gamma + O\left( \frac{d\beta_T}{\Gamma} \log \left( 1 + \frac{L^2 \beta_T}{\lambda \Gamma^2} \right) \right) + T\Gamma + O\left( \frac{d\beta_T \log T}{\Gamma} \log \left( 1 + \frac{L^2 \beta_T \log T}{\lambda \Gamma^2} \right) \right)$$

$$+ T\Gamma + O\left( \frac{d\beta_T}{\Gamma} \log \left( 1 + \frac{L^2 \beta_T}{\lambda \Gamma^2} \right) \right) + T\Gamma + O\left( \frac{d\beta_T \log T}{\Gamma} \log \left( 1 + \frac{L^2 \beta_T \log T}{\lambda \Gamma^2} \right) \right)$$

$$+ T\Gamma + O\left( \sqrt{T\beta_T \log T} \log \left( \frac{L^2 \beta_T \log T}{\lambda \Gamma^2} \right) \right)$$

$$\leq 5T\Gamma + O\left( \frac{d\beta_T \log T}{\Gamma} \log \left( 1 + \frac{L^2 \beta_T \log T}{\lambda \Gamma^2} \right) \right) + O\left( \sqrt{T \log T} \log \left( \frac{L^2 \beta_T \log T}{\lambda \Gamma^2} \right) \right) .$$

□

### D.5 Proof of Lemma 9

*Proof.* The proof of this case is straightforward by using Lemma 1 with the choice $\gamma = \frac{1}{t^2}$:

$$F_4 = \mathbb{E} \sum_{t=1}^{T} \Delta_t \cdot \mathbb{1}\left\{ \overline{B}_t \right\}$$

$$= \mathbb{E} \sum_{t=1}^{T} \Delta_t \cdot \mathbb{1}\left\{ \overline{B}_t, \Delta_t < \Gamma \right\} + \mathbb{E} \sum_{t=1}^{T} \Delta_t \cdot \mathbb{1}\left\{ \overline{B}_t, \Delta_t \geq \Gamma \right\}$$

$$< T\Gamma + \mathbb{E} \sum_{t=1}^{T} \sum_{l=1}^{\lceil Q \rceil} \Delta_t \cdot \mathbb{1}\left\{ \overline{B}_t, 2^{-l} < \Delta_t \leq 2^{-l+1} \right\}$$

$$\leq T\Gamma + \mathbb{E} \sum_{t=1}^{T} \sum_{l=1}^{\lceil Q \rceil} 2^{-l+1} \cdot \mathbb{1}\left\{ \overline{B}_t \right\}$$

$$\leq T\Gamma + \sum_{l=1}^{\lceil Q \rceil} 2^{-l+1} \sum_{t=1}^{T} \mathbb{P}\left\{ \overline{B}_t \right\}$$

$$= T\Gamma + \sum_{l=1}^{\lceil Q \rceil} 2^{-l+1} \cdot \frac{\pi^2}{6}$$

$$< T\Gamma + (2 - \Gamma) \cdot \frac{\pi^2}{6}$$

$$< T\Gamma + \frac{\pi^2}{3}$$

$$= T\Gamma + O(1) .$$

□

### D.6 Proof of Theorem 2

*Proof.* Combining Lemmas 6, 7, 8 and 9, with the choices of $\gamma$ and $\Gamma$ as in Eqn. (21), the regret of LinIMED-2 is bounded as follows:

$$R_T = F_1 + F_2 + F_3 + F_4$$

$$\leq T\Gamma + O\left(\frac{d\beta_T \log T}{\Gamma}\right)\log\left(1 + \frac{L^2\beta_T \log T}{\lambda\Gamma^2}\right) + T\Gamma + O\left(\frac{d\beta_T \log T}{\Gamma}\right)\log\left(1 + \frac{L^2\beta_T \log T}{\lambda\Gamma^2}\right)$$

$$+ 5T\Gamma + O\left(\frac{d\beta_T \log T}{\Gamma}\log\left(1 + \frac{L^2\beta_T \log T}{\lambda\Gamma^2}\right)\right) + O\left(\sqrt{T\log T}\log\left(\frac{L^2\beta_T \log T}{\lambda\Gamma^2}\right)\right)$$

$$+ T\Gamma + O(1)$$

$$\leq 8T\Gamma + O\left(\frac{d\beta_T \log T}{\Gamma}\log\left(1 + \frac{L^2\beta_T \log T}{\lambda\Gamma^2}\right)\right) + O\left(\sqrt{T\log T}\log\left(\frac{L^2\beta_T \log T}{\lambda\Gamma^2}\right)\right)$$

$$= 8\sqrt{dT\beta_T}\log T + O\left(\sqrt{dT\beta_T}\log\left(1 + \frac{TL^2}{\lambda d\log T}\right)\right) + O\left(\sqrt{T\log T}\log\left(\frac{TL^2}{\lambda d\log T}\right)\right)$$

$$= 8d\sqrt{T}\log^{\frac{3}{2}} T + O\left(d\sqrt{T}\log^{\frac{3}{2}} T\right) + O\left(\sqrt{T}\log^{\frac{3}{2}} T\right)$$

$$\leq O\left(d\sqrt{T}\log^{\frac{3}{2}} T\right).$$

$\square$

# E  Proof of the regret bound for LinIMED-3 (Proof of Theorem 3)

First we define $a_t^*$ as the best arm in time step $t$ such that $a_t^* = \arg\max_{a \in \mathcal{A}_t}\langle\theta^*, x_{t,a}\rangle$, and use $x_t^* := x_{t,a_t^*}$ denote its corresponding context. Define $\hat{A}_t := \arg\max_{a \in \mathcal{A}_t}\mathrm{UCB}_t(a)$. Let $\Delta_t := \langle\theta^*, x_t^*\rangle - \langle\theta^*, X_t\rangle$ denote the regret in time $t$. Define the following events:

$$B_t' := \left\{\|\hat{\theta}_{t-1} - \theta^*\|_{V_{t-1}} \leq \sqrt{\beta_{t-1}(\gamma)}\right\}, \quad D_t' := \left\{\hat{\Delta}_{t,A_t} > \varepsilon\right\}.$$

where $\varepsilon$ is a free parameter set to be $\varepsilon = \frac{\Delta_t}{3}$ in this proof sketch.

Then the expected regret $R_T = \mathbb{E}\sum_{t=1}^T \Delta_t$ can be partitioned by events $B_t', D_t'$ such that:

$$R_T = \underbrace{\mathbb{E}\sum_{t=1}^T \Delta_t \cdot \mathbb{1}\left\{B_t', D_t'\right\}}_{=:F_1} + \underbrace{\mathbb{E}\sum_{t=1}^T \Delta_t \cdot \mathbb{1}\left\{B_t, \overline{D_t'}\right\}}_{=:F_2} + \underbrace{\mathbb{E}\sum_{t=1}^T \Delta_t \cdot \mathbb{1}\left\{\overline{B_t'}\right\}}_{=:F_3}.$$

**For the $F_1$ case:**
From $D_t'$ we know $A_t \neq \hat{A}_t$, therefore

$$I_{t,A_t} = \frac{\hat{\Delta}_{t,A_t}^2}{\beta_{t-1}\|X_t\|_{V_{t-1}^{-1}}^2} + \log\frac{1}{\beta_{t-1}\|X_t\|_{V_{t-1}^{-1}}^2}. \tag{29}$$

From $D_t'$ and $I_{t,A_t} \leq I_{t,\hat{A}_t} \leq \log\frac{C}{\max_{a\in\mathcal{A}_t}\hat{\Delta}_{t,a}^2}$, we have

$$I_{t,A_t} < \log\frac{C}{\varepsilon^2}. \tag{30}$$

Combining Eqn. (29) and Eqn. (30),

$$\frac{\hat{\Delta}_{t,A_t}^2}{\beta_{t-1}\|X_t\|_{V_{t-1}^{-1}}^2} + \log\frac{1}{\beta_{t-1}\|X_t\|_{V_{t-1}^{-1}}^2} < \log\frac{C}{\varepsilon^2}.$$

Then

$$\frac{\hat{\Delta}_{t,A_t}^2}{\beta_{t-1}\|X_t\|_{V_{t-1}^{-1}}^2} < \log\left(\beta_{t-1}\|X_t\|_{V_{t-1}^{-1}}^2 \cdot \frac{C}{\varepsilon^2}\right). \tag{31}$$

If $\|X_t\|^2_{V^{-1}_{t-1}} \geq \frac{\Delta^2_t}{\beta_{t-1}}$, using the same procedure from Eqn. (15) to Eqn. (16), one has:

$$\mathbb{E}\sum_{t=1}^{T}\Delta_t \cdot \mathbb{1}\{B'_t, D'_t\} \cdot \mathbb{1}\left\{\|X_t\|^2_{V^{-1}_{t-1}} \geq \frac{\Delta^2_t}{\beta_{t-1}}\right\}$$

$$\leq \mathbb{E}\sum_{t=1}^{T}\Delta_t \cdot \mathbb{1}\left\{\|X_t\|^2_{V^{-1}_{t-1}} \geq \frac{\Delta^2_t}{\beta_{t-1}}\right\}$$

$$< T\Gamma + \frac{48d\beta_T}{\Gamma}\log\left(1 + \frac{8L^2\beta_T}{\lambda\Gamma^2}\right)$$

$$= T\Gamma + O\left(\frac{d\beta_T}{\Gamma}\log\left(1 + \frac{L^2\beta_T}{\lambda\Gamma^2}\right)\right).$$

Else $\|X_t\|^2_{V^{-1}_{t-1}} < \frac{\Delta^2_t}{\beta_{t-1}}$, this implies that $\beta_{t-1}\|X_t\|^2_{V^{-1}_{t-1}} < \Delta^2_t$, plug this into Eqn. (31) and with the choice of $\varepsilon = \frac{\Delta_t}{3}$ and $D'_t$, we have

$$\frac{\Delta^2_t}{9\beta_{t-1}\|X_t\|^2_{V^{-1}_{t-1}}} < \log(9C) .$$

Since $C \geq 1$ is a constant, then

$$\|X_t\|^2_{V^{-1}_{t-1}} > \frac{\Delta^2_t}{9\beta_{t-1}\log(9C)} .$$

Using the same procedure from Eqn. (15) to Eqn. (16), one has:

$$\mathbb{E}\sum_{t=1}^{T}\Delta_t \cdot \mathbb{1}\{B'_t, D'_t\} \cdot \mathbb{1}\left\{\|X_t\|^2_{V^{-1}_{t-1}} < \frac{\Delta^2_t}{\beta_{t-1}}\right\}$$

$$\leq \mathbb{E}\sum_{t=1}^{T}\Delta_t \cdot \mathbb{1}\left\{\|X_t\|^2_{V^{-1}_{t-1}} > \frac{\Delta^2_t}{9\beta_{t-1}\log(9C)}\right\}$$

$$< T\Gamma + O\left(\frac{d\beta_T\log C}{\Gamma}\log\left(1 + \frac{L^2\beta_T\log C}{\lambda\Gamma^2}\right)\right) .$$

Hence

$$F_1 < 2T\Gamma + O\left(\frac{d\beta_T\log C}{\Gamma}\log\left(1 + \frac{L^2\beta_T\log C}{\lambda\Gamma^2}\right)\right) . \tag{32}$$

**For the $F_2$ case:** Since the event $B'_t$ holds,

$$\max_{a\in\mathcal{A}_t}\text{UCB}_t(a) \geq \text{UCB}_t(a^*_t) = \langle\hat{\theta}_{t-1}, x^*_t\rangle + \sqrt{\beta_{t-1}}\|x^*_t\|_{V^{-1}_{t-1}} \geq \langle\theta^*, x^*_t\rangle \tag{33}$$

On the other hand, from $\overline{D'_t}$ we have

$$\max_{a\in\mathcal{A}_t}\text{UCB}_t(a) \leq \text{UCB}_t(A_t) + \varepsilon = \langle\hat{\theta}_{t-1}, X_t\rangle + \sqrt{\beta_{t-1}}\|X_t\|_{V^{-1}_{t-1}} + \varepsilon . \tag{34}$$

Combining Eqn. (33) and Eqn. (34),

$$\langle\theta^*, x^*_t\rangle \leq \langle\hat{\theta}_{t-1}, X_t\rangle + \sqrt{\beta_{t-1}}\|X_t\|_{V^{-1}_{t-1}} + \varepsilon .$$

Hence

$$\Delta_t - \varepsilon \leq \langle\hat{\theta}_{t-1} - \theta^*, X_t\rangle + \sqrt{\beta_{t-1}}\|X_t\|_{V^{-1}_{t-1}} .$$

Then with $\varepsilon = \frac{\Delta_t}{3}$ and $B'_t$, we have

$$\frac{2}{3}\Delta_t \leq 2\sqrt{\beta_{t-1}}\|X_t\|_{V_{t-1}^{-1}} \,,$$

therefore

$$\|X_t\|_{V_{t-1}^{-1}}^2 > \frac{\Delta_t^2}{9\beta_{t-1}} \,.$$

Using the same procedure from Eqn. (15) to Eqn. (16), one has:

$$F_2 < T\Gamma + O\left(\frac{d\beta_T}{\Gamma}\log\left(1 + \frac{L^2\beta_T}{\lambda\Gamma^2}\right)\right) \,. \tag{35}$$

**For the $F_3$ case:**
using Lemma 1 with the choice $\gamma = \frac{1}{t^2}$:

$$
\begin{aligned}
F_3 &= \mathbb{E}\sum_{t=1}^{T}\Delta_t \cdot \mathbb{1}\left\{\overline{B'_t}\right\} \\
&= \mathbb{E}\sum_{t=1}^{T}\Delta_t \cdot \mathbb{1}\left\{\overline{B'_t}, \Delta_t < \Gamma\right\} + \mathbb{E}\sum_{t=1}^{T}\Delta_t \cdot \mathbb{1}\left\{\overline{B'_t}, \Delta_t \geq \Gamma\right\} \\
&< T\Gamma + \mathbb{E}\sum_{t=1}^{T}\sum_{l=1}^{\lceil Q\rceil}\Delta_t \cdot \mathbb{1}\left\{\overline{B'_t}, 2^{-l} < \Delta_t \leq 2^{-l+1}\right\} \\
&\leq T\Gamma + \mathbb{E}\sum_{t=1}^{T}\sum_{l=1}^{\lceil Q\rceil}2^{-l+1} \cdot \mathbb{1}\left\{\overline{B'_t}\right\} \\
&\leq T\Gamma + \sum_{l=1}^{\lceil Q\rceil}2^{-l+1}\sum_{t=1}^{T}\mathbb{P}\left\{\overline{B'_t}\right\} \\
&= T\Gamma + \sum_{l=1}^{\lceil Q\rceil}2^{-l+1} \cdot \frac{\pi^2}{6} \\
&< T\Gamma + (2 - \Gamma) \cdot \frac{\pi^2}{6} \\
&< T\Gamma + \frac{\pi^2}{3} \\
&= T\Gamma + O(1) \,. \tag{36}
\end{aligned}
$$

### E.1 Proof of Theorem 3

*Proof.* Combining Eqn. (32), (35), (36) with the choices of $\gamma = \frac{1}{t^2}$ and $\Gamma = \frac{\beta_T}{\sqrt{T}}$ and $C \geq 1$ is a constant, the regret of LinIMED-3 is bounded as follows:

$$
\begin{aligned}
R_T &= F_1 + F_2 + F_3 + F_4 \\
&< 4T\Gamma + O\left(\frac{d\beta_T \log C}{\Gamma}\log\left(1 + \frac{L^2\beta_T \log C}{\lambda\Gamma^2}\right)\right) + O\left(\frac{d\beta_T}{\Gamma}\log\left(1 + \frac{L^2\beta_T}{\lambda\Gamma^2}\right)\right) + O(1) \\
&< O\left(d\sqrt{T}\log C\log\left(1 + \frac{L^2 T \log C}{\lambda}\right)\right) \\
&= O\left(d\sqrt{T}\log(T)\right) \,.
\end{aligned}
$$

This completes the proof. $\qquad\square$

# F   Proof of the regret bound for SupLinIMED (Proof of Theorem 4)

Define $s_t \in [[\log T]]$ as the index of $s$ when the arm is chosen at time $t$. For the SupLinIMED, the index of arms except the empirically best arm is defined by $I_{t,a} = \left( \frac{\hat{\Delta}_{t,a}^{s_t}}{w_{t,a}^{s_t}} \right)^2 - 2\log(w_{t,a}^{s_t})$, whereas the index of the empirically best arm is defined by $I_{t,\hat{A}_t^*} = \log(2T) \wedge (-2\log(w_{t,\hat{A}_t^*}^{s_t}))$ where $\hat{A}_t^* = \arg\max_{a \in \hat{A}_{s_t}} \langle \hat{\theta}_t^{s_t}, x_{t,a} \rangle$. Define the index of the best arm at time $t$ as $a_t^* := \arg\max_{a \in [K]} \langle \theta^*, x_{t,a} \rangle$.

**Remark 1.** *Here the upper bound we set for the index of the empirically best arm is $\log(2T)$, which is slightly larger than our previous $\log T$ (Line 10 in the LinIMED algorithm) since in the first step of the of the SupLinIMED algorithm or, more generally, the SupLinUCB-type algorithms, the width of each arm is less than $\frac{1}{\sqrt{T}}$, as a result, the index of each arm is larger than $\log T$.*

Let the set of time indices such that the chosen arm is from Step 1 (Lines 6–9 in Algorithm 2) be $\Psi_0$. Then the cumulative expected regret of the SupLinIMED algorithm over time horizon $T$ can be defined by the following equation:

$$R_T = \mathbb{E}\left[ \sum_{t \in \Psi_0} \langle \theta^*, x_{t,a_t^*} - X_t \rangle \right] + \mathbb{E}\left[ \sum_{t \notin \Psi_0} \langle \theta^*, x_{t,a_t^*} - X_t \rangle \right] \tag{37}$$

Since the index set has not changed in Step 1 (see Line 9 in Algorithm 2), the second term of the regret is the same as in the original SupLinUCB algorithm of Chu et al. (2011). For the first term, we partitioned it by the following events:

$$B_t := \bigcap_{t \in [T], s \in [\log T], a \in [K]} \left\{ |\langle \theta^* - \hat{\theta}_t^s, x_{t,a} \rangle| \leq \frac{\alpha + 1}{\alpha} w_{t,a}^s \right\}, \qquad \text{and}$$

$$D_t := \left\{ \hat{\Delta}_{t,A_t}^{s_t} \geq \varepsilon \right\},$$

where $\alpha = \sqrt{\frac{1}{2} \ln \frac{2TK}{\gamma}}$ as in the SupLinUCB (Chu et al., 2011). We choose $\gamma = \frac{1}{2t^2}$ throughout. Furthermore, $\hat{\theta}_t^s$ is the $\hat{\theta}_t$ obtained from Algorithm 3 with $\Psi_t^s$ as the input, i.e.,

$$\hat{\theta}_t^s := \left( I_d + \sum_{\tau \in \Psi_t^s} x_{\tau,A_\tau} x_{\tau,A_\tau}^T \right)^{-1} \sum_{\tau \in \Psi_t^s} Y_{\tau,A_\tau} x_{\tau,A_\tau}.$$

Define $\Delta_t := \langle \theta^*, x_{t,a_t^*} - X_t \rangle$ as the instantaneous regret at each time step $t$. In addition, choose $\varepsilon = \frac{\Delta_t}{3}$ in the definition of $D_t$. Then the first term of the expected regret in (37) can be partitioned by the events $B_t$ and $D_t$ as follows:

$$\mathbb{E}\left[ \sum_{t \in \Psi_0} \langle \theta^*, x_{t,a_t^*} - X_t \rangle \right] = \underbrace{\mathbb{E}\left[ \sum_{t \in \Psi_0} \Delta_t \cdot \mathbb{1}\{B_t, D_t\} \right]}_{=:F_1} + \underbrace{\mathbb{E}\left[ \sum_{t \in \Psi_0} \Delta_t \cdot \mathbb{1}\{B_t, \overline{D}_t\} \right]}_{=:F_2} + \underbrace{\mathbb{E}\left[ \sum_{t \in \Psi_0} \Delta_t \cdot \mathbb{1}\{\overline{B}_t\} \right]}_{=:F_3}$$

We recall that when $t \in \Psi_0$, $w_{t,a}^{s_t} \leq \frac{1}{\sqrt{T}}$ for all $a \in \hat{A}_{s_t}$.

To bound $F_1$, we note that since $B_t$ occurs, the actual best arm $a_t^* \in \hat{A}_{s_t}$ with high probability $(1 - \gamma \log^2 T)$ by Chu et al. (2011, Lemma 5). As such,

$$\max_{a \in \hat{A}_{s_t}} \langle \hat{\theta}_t^{s_t}, x_{t,a} \rangle \geq \langle \hat{\theta}_t^{s_t}, x_{t,a_t^*} \rangle \geq \langle \theta^*, x_{t,a_t^*} \rangle - \frac{\alpha + 1}{\alpha} w_{t,a_t^*}^s \geq \langle \theta^*, x_{t,a_t^*} \rangle - \frac{2}{\sqrt{T}}$$

where the last inequality is from the fact that $\gamma = \frac{1}{2t^2}$ and $\alpha = \sqrt{\frac{1}{2}\ln\frac{2TK}{\gamma}} \geq 1$ . Else if, in fact, the best arm $a_t^* \notin \hat{\mathcal{A}}_{s_t}$, the corresponding regret in this case is bounded by:

$$\mathbb{E}\sum_{t\in\Psi_0}\Delta_t \cdot \mathbb{1}\left\{a_t^* \notin \hat{\mathcal{A}}_{s_t}\right\} \leq \mathbb{E}\sum_{t=1}^{T}\Delta_t \cdot \mathbb{1}\left\{a_t^* \notin \hat{\mathcal{A}}_{s_t}\right\}$$

$$\leq \mathbb{E}\sum_{t=1}^{T}\mathbb{1}\left\{a_t^* \notin \hat{\mathcal{A}}_{s_t}\right\}$$

$$= \sum_{t=1}^{T}\mathbb{P}(a_t^* \notin \hat{\mathcal{A}}_{s_t})$$

$$\leq \sum_{t=1}^{T}\gamma\log^2 T$$

$$= \sum_{t=1}^{T}\frac{\log^2 T}{2t^2}$$

$$< \frac{\pi^2}{12}\log^2 T .$$

**Case 1:** If $\hat{A}_t^* \neq A_t$, this means that the index of $A_t$ is $I_{t,A_t} = \frac{(\hat{\Delta}_{t,A_t}^{s_t})^2}{\alpha^2\|X_t\|_{V_t^{-1}}^2} + \log\frac{1}{\alpha^2\|X_t\|_{V_t^{-1}}^2}$. Using the fact that $I_{t,A_t} \leq I_{t,\hat{A}_t^*}$ we have

$$\frac{(\hat{\Delta}_{t,A_t}^{s_t})^2}{\alpha^2\|X_t\|_{V_t^{-1}}^2} + \log\frac{1}{\alpha^2\|X_t\|_{V_t^{-1}}^2} \leq 2\log T .$$

Then using the definition of the event $D_t$ and the fact that $(w_{t,a}^{s_t})^2 = \alpha^2\|X_t\|_{V_t^{-1}}^2 \leq \frac{1}{T}$, we have

$$\Delta_t^2 \leq 9\alpha^2\|X_t\|_{V_{t-1}^{-1}}^2\log T \leq \frac{9\log T}{T}.$$

Hence, $\Delta_t \leq \frac{3\sqrt{\log T}}{\sqrt{T}}$. Therefore $F_1$ in this case is upper bounded as follows:

$$\mathbb{E}\left[\sum_{t\in\Psi_0}\Delta_t \cdot \mathbb{1}\{B_t, D_t\} \cdot \mathbb{1}\left\{\hat{A}_t^* \neq A_t\right\} \cdot \mathbb{1}\left\{a_t^* \in \hat{\mathcal{A}}_{s_t}\right\}\right] \leq \mathbb{E}\left[\sum_{t\in\Psi_0}\frac{3\sqrt{\log T}}{\sqrt{T}}\right] \leq 3\sqrt{T\log T} .$$

**Case 2:** If $\hat{A}_t^* = A_t$, then using the definition of the event $B_t$, we have

$$\langle\hat{\theta}_t^{s_t}, X_t\rangle = \max_{a\in\hat{\mathcal{A}}_{s_t}}\langle\hat{\theta}_t^{s_t}, x_{t,a}\rangle \geq \langle\hat{\theta}_t^{s_t}, x_{t,a_t^*}\rangle \geq \langle\theta^*, x_{t,a_t^*}\rangle - \frac{2}{\sqrt{T}} = \langle\theta^*, X_t\rangle + \Delta_t - \frac{2}{\sqrt{T}}$$

therefore since event $B_t$ occurs,

$$\Delta_t \leq \langle\hat{\theta}_t^{s_t} - \theta^*, X_t\rangle + \frac{2}{\sqrt{T}} \leq \frac{3}{\sqrt{T}} .$$

Hence $F_1$ in this case is bounded as $2\sqrt{T}$. Combining the above cases,

$$F_1 \leq 3\sqrt{T\log T} + 3\sqrt{T} + \frac{\pi^2}{12}\log^2 T \leq O(\sqrt{T\log T}) .$$

To bound $F_2$, we note from the definition of $B_t$ that

$$\max_{a \in \hat{\mathcal{A}}_{s_t}} \langle \hat{\theta}_t^{s_t}, x_{t,a} \rangle \geq \langle \hat{\theta}_t^{s_t}, x_{t,a_t^*} \rangle \geq \langle \theta^*, x_{t,a_t^*} \rangle - \frac{2}{\sqrt{T}}$$

then on the event $\overline{D}_t$,

$$\langle \theta^*, x_{t,a_t^*} \rangle - \frac{2}{\sqrt{T}} \leq \max_{a \in \hat{\mathcal{A}}_{s_t}} \langle \hat{\theta}_t^{s_t}, x_{t,a} \rangle < \varepsilon + \langle \hat{\theta}_t^{s_t}, X_t \rangle = \frac{\Delta_t}{3} + \langle \hat{\theta}_t^{s_t}, X_t \rangle ,$$

therefore

$$\Delta_t < \frac{3}{2} \left( \langle \hat{\theta}_t^{s_t} - \theta^*, X_t \rangle + \frac{2}{\sqrt{T}} \right) \leq \frac{9}{2\sqrt{T}}$$

Hence

$$\begin{aligned}
F_2 &= \mathbb{E}\left[ \sum_{t \in \Psi_0} \Delta_t \cdot \mathbb{1}\left\{ B_t, \overline{D}_t \right\} \cdot \mathbb{1}\left\{ a_t^* \in \hat{\mathcal{A}}_{s_t} \right\} \right] + \mathbb{E}\left[ \sum_{t \in \Psi_0} \Delta_t \cdot \mathbb{1}\left\{ B_t, \overline{D}_t \right\} \cdot \mathbb{1}\left\{ a_t^* \notin \hat{\mathcal{A}}_{s_t} \right\} \right] \\
&\leq \mathbb{E}\left[ \sum_{t=1}^{T} \Delta_t \cdot \mathbb{1}\left\{ B_t, \overline{D}_t \right\} \right] + \frac{\pi^2}{12} \log^2 T \\
&< \mathbb{E}\left[ \sum_{t=1}^{T} \frac{9}{2\sqrt{T}} \cdot \mathbb{1}\left\{ B_t, \overline{D}_t \right\} \right] + \frac{\pi^2}{12} \log^2 T \\
&< T \cdot \frac{9}{2\sqrt{T}} + \frac{\pi^2}{12} \log^2 T \\
&= \frac{9}{2}\sqrt{T} + \frac{\pi^2}{12} \log^2 T \\
&\leq O(\sqrt{T}) .
\end{aligned}$$

To bound $F_3$, we use the proof as in of Chu et al. (2011, Lemma 1), which is restated as follows.

**Lemma 10.** *For any $a \in [K]$, $s \in [\lceil \log T \rceil]$, $t \in [T]$,*

$$\mathbb{P}\left[ |\langle \theta^* - \hat{\theta}_t^s, x_{t,a} \rangle| > \frac{\alpha + 1}{\alpha} w_{t,a}^s \right] \leq \frac{\gamma}{TK}$$

*where $\alpha = \sqrt{\frac{1}{2} \ln \frac{2TK}{\gamma}}$.*

Then using the union bound, we have for all $t \in [T]$, $s \in [\lceil \log T \rceil]$, for all $a \in [K]$,

$$\begin{aligned}
\mathbb{P}\left[ \overline{B}_t \right] &= \mathbb{P}\left[ \bigcup_{t \in [T], s \in [\lceil \log T \rceil], a \in [K]} \left\{ |\langle \theta^* - \hat{\theta}_t^s, x_{t,a} \rangle| > \frac{\alpha + 1}{\alpha} w_{t,a}^s \right\} \right] \\
&\leq \sum_{t \in [T]} \sum_{s \in [\lceil \log T \rceil]} \sum_{a \in [K]} \mathbb{P}\left[ |\langle \theta^* - \hat{\theta}_t^s, x_{t,a} \rangle| > \frac{\alpha + 1}{\alpha} w_{t,a}^s \right] \\
&< \left( TK(1 + \log T) \right) \frac{\gamma}{TK} \\
&= \gamma(1 + \log T) .
\end{aligned}$$

With the choice $\gamma = \frac{1}{2t^2}$ and the assumption $\Delta_t \leq 1$,

$$
\begin{aligned}
F_3 &= \mathbb{E}\left[\sum_{t \in \Psi_0} \Delta_t \cdot \mathbb{1}\left\{\overline{B}_t\right\}\right] \\
&\leq \sum_{t=1}^{T} \mathbb{P}\left[\overline{B}_t\right] \\
&< \sum_{t=1}^{T} \frac{1 + \log T}{2t^2} \\
&< \frac{\pi^2}{12}(1 + \log T) \\
&\leq O(\log T) \ .
\end{aligned}
$$

Hence the first term in $R_T$ in (37) is upper bounded by:

$$
\begin{aligned}
\mathbb{E}\left[\sum_{t \in \Psi_0} \langle \theta^*, x_{t,a_t^*} - X_t \rangle\right] &\leq O(\sqrt{T}) + O(\log T) + O(\sqrt{T \log T}) \\
&\leq O(\sqrt{T \log T}) \ .
\end{aligned}
$$

On the other hand, by Chu et al. (2011, Theorem 1), the second term in $R_T$ in (37) is upper bounded as follows:

$$
\mathbb{E}\left[\sum_{t \notin \Psi_0} \langle \theta^*, x_{t,a_t^*} - X_t \rangle\right] \leq O\left(\sqrt{dT \log^3(KT)}\right).
$$

Hence the regret of our algorithm SupLinIMED is upper bounded as follows:

$$
R_T \leq O\left(\sqrt{dT \log^3(KT)}\right) \ .
$$

This completes the proof of Theorem 4.

## G   Hyperparameter tuning in our empirical study

### G.1   Synthetic Dataset

The below tables are the empirical results while tuning the hyperparameter $\alpha$ (scale of the confidence width) for fixed $T = 1000$.

| Method | LinUCB | | | LinTS | | | LinIMED-1 | | | LinIMED-2 | | | LinIMED-3 ($C = 30$) | | |
|---|---|---|---|---|---|---|---|---|---|---|---|---|---|---|---|
| $\alpha$ | 0.5 | 0.55 | 0.6 | 0.2 | 0.25 | 0.3 | 0.15 | 0.2 | 0.25 | 0.2 | 0.25 | 0.3 | 0.15 | 0.2 | 0.25 |
| Regret | 7.780 | 6.695 | 6.856 | 9.769 | 9.201 | 12.068 | 24.086 | 5.482 | 6.108 | 4.999 | 4.998 | 7.329 | 25.588 | 2.075 | 2.760 |

Table 2: Tuning $\alpha$ when $K = 10, d = 2$

| Method | LinUCB | | | LinTS | | | LinIMED-1 | | | LinIMED-2 | | | LinIMED-3 ($C = 30$) | | |
|---|---|---|---|---|---|---|---|---|---|---|---|---|---|---|---|
| $\alpha$ | 0.5 | 0.55 | 0.6 | 0.1 | 0.15 | 0.2 | 0.2 | 0.25 | 0.3 | 0.2 | 0.25 | 0.3 | 0.2 | 0.25 | 0.3 |
| Regret | 7.203 | 6.832 | 7.423 | 54.221 | 7.042 | 7.352 | 6.707 | 6.053 | 8.458 | 6.254 | 4.918 | 7.013 | 4.407 | 2.562 | 3.041 |

Table 3: Tuning $\alpha$ when $K = 100, d = 2$

| Method | LinUCB | | | LinTS | | | LinIMED-1 | | | LinIMED-2 | | | LinIMED-3 ($C = 30$) | | |
|---|---|---|---|---|---|---|---|---|---|---|---|---|---|---|---|
| $\alpha$ | 0.5 | 0.55 | 0.6 | 0.1 | 0.15 | 0.2 | 0.15 | 0.2 | 0.25 | 0.2 | 0.25 | 0.3 | 0.15 | 0.2 | 0.25 |
| Regret | 7.919 | 5.679 | 7.063 | 69.955 | 6.925 | 7.037 | 24.393 | 5.625 | 6.335 | 6.335 | 4.831 | 7.040 | 41.355 | 1.936 | 2.250 |

Table 4: Tuning $\alpha$ when $K = 500, d = 2$

| Method | LinUCB | | | LinTS | | | LinIMED-1 | | | LinIMED-2 | | | LinIMED-3 ($C = 30$) | | |
|---|---|---|---|---|---|---|---|---|---|---|---|---|---|---|---|
| $\alpha$ | 0.45 | 0.5 | 0.55 | 0.1 | 0.15 | 0.2 | 0.1 | 0.15 | 0.2 | 0.1 | 0.15 | 0.2 | 0.1 | 0.15 | 0.2 |
| Regret | 9.164 | 9.094 | 14.183 | 14.252 | 9.886 | 14.680 | 19.663 | 6.463 | 10.643 | 15.685 | 5.399 | 8.373 | 8.024 | 2.062 | 3.342 |

Table 5: Tuning $\alpha$ when $K = 10, d = 20$

| Method | LinUCB | | | LinTS | | | LinIMED-1 | | | LinIMED-2 | | | LinIMED-3 ($C = 30$) | | |
|---|---|---|---|---|---|---|---|---|---|---|---|---|---|---|---|
| $\alpha$ | 0.25 | 0.3 | 0.35 | 0.1 | 0.15 | 0.2 | 0.05 | 0.1 | 0.15 | 0.1 | 0.15 | 0.2 | 0.05 | 0.1 | 0.15 |
| Regret | 7.923 | 7.085 | 10.981 | 14.983 | 9.565 | 19.300 | 58.278 | 6.165 | 9.225 | 8.916 | 8.575 | 13.483 | 142.704 | 2.816 | 3.497 |

Table 6: Tuning $\alpha$ when $K = 10, d = 50$

We run these algorithms on the same dataset with different choices of $\alpha$, we choose the best $\alpha$ with the corresponding least regret.

## G.2 MovieLens Dataset

The below tables are the empirical results while tuning the hyperparameter $\alpha$ (scale of the confidence width) for fixed $T = 1000$.

| Method | LinUCB | | | LinTS | | | LinIMED-1 | | | LinIMED-2 | | | LinIMED-3 ($C = 30$) | | | IDS | | |
|---|---|---|---|---|---|---|---|---|---|---|---|---|---|---|---|---|---|---|
| $\alpha$ | 0.7 | 0.75 | 0.8 | 0.05 | 0.1 | 0.15 | 0.15 | 0.2 | 0.25 | 0.15 | 0.2 | 0.25 | 0.2 | 0.25 | 0.3 | 0.25 | 0.3 | 0.35 |
| CTR | 0.608 | 0.675 | 0.668 | 0.615 | 0.705 | 0.679 | 0.740 | 0.823 | 0.766 | 0.740 | 0.823 | 0.766 | 0.713 | 0.742 | 0.690 | 0.655 | 0.728 | 0.714 |

Table 7: Tuning $\alpha$ when $K = 20$

| Method | LinUCB | | | LinTS | | | LinIMED-1 | | | LinIMED-2 | | | LinIMED-3 ($C = 30$) | | | IDS | | |
|---|---|---|---|---|---|---|---|---|---|---|---|---|---|---|---|---|---|---|
| $\alpha$ | 0.75 | 0.8 | 0.85 | 0 | 0.05 | 0.1 | 0.1 | 0.15 | 0.2 | 0.05 | 0.1 | 0.15 | 0.05 | 0.1 | 0.15 | 0.3 | 0.35 | 0.4 |
| CTR | 0.708 | 0.754 | 0.713 | 0.517 | 0.711 | 0.646 | 0.648 | 0.668 | 0.595 | 0.658 | 0.668 | 0.651 | 0.697 | 0.717 | 0.649 | 0.643 | 0.688 | 0.606 |

Table 8: Tuning $\alpha$ when $K = 50$

| Method | LinUCB | | | LinTS | | | LinIMED-1 | | | LinIMED-2 | | | LinIMED-3 ($C = 30$) | | | IDS | | |
|---|---|---|---|---|---|---|---|---|---|---|---|---|---|---|---|---|---|---|
| $\alpha$ | 0.85 | 0.9 | 0.95 | 0 | 0.05 | 0.1 | 0.05 | 0.1 | 0.15 | 0.05 | 0.1 | 0.15 | 0.05 | 0.1 | 0.15 | 0.3 | 0.35 | 0.4 |
| CTR | 0.721 | 0.754 | 0.745 | 0.487 | 0.674 | 0.588 | 0.682 | 0.729 | 0.594 | 0.687 | 0.729 | 0.594 | 0.689 | 0.705 | 0.594 | 0.684 | 0.739 | 0.695 |

Table 9: Tuning $\alpha$ when $K = 100$

We run these algorithms on the same dataset with different choices of $\alpha$ and we choose the best $\alpha$ with the corresponding largest reward.

