# OpenReview forum: "Indexed Minimum Empirical Divergence-Based Algorithms for Linear Bandits"
_TMLR — Accepted by TMLR_

### Review · Reviewer_CTEL · 2024-02-13

**Summary Of Contributions:**

This paper considers the extension of the Indexed Minimum Empirical Divergence (IMED) family of algorithms (Honda and Takemura, 2015) to the linear bandit problem. The paper proposes three variants of linear IMED with different parameterisation of its indices and a SupLinIMED algorithm, inspired by the SupLinUCB/SupLinTS family of algorithms.

The motivation for proposing these algorithms is that existing algorithms for linear bandit problems are either computationally intensive or carry suboptimal theoretical guarantees. Inspired by the success of IMED algorithms for standard bandit problems, the linear IMED algorithms proposed realise closer to optimal order regret than is attainable with randomised policies with improved computational efficiency over other near-optimal algorithms.

The paper derives guarantees on the regret of all four proposed algorithms and studies all but the SupLinIMED algorithm in three numerical experiments: two synthetic and one constructed on the MovieLens dataset. The performance of the third algorithm LinIMED-3 is seen to be the strongest across these experiments.

**Audience:**

Yes

**Broader Impact Concerns:**

I do not have broader impact concerns for this principally theoretical work.

**Claims And Evidence:**

Yes

**Requested Changes:**

I feel the scope of the paper and the theoretical work are good and would upon resolution of concerns lean towards acceptance. However I would ideally have clarification on points 1-9 above and on the following additional remarks which are arranged in important (I) and strengthening (S) categories:

I1: What is the dependence of the theoretical results on S, L, and R? And why is Assumption 1 bounding the gaps with 1 and not a free parameter like the other bounds? Does the assumption 1 bound imply something on S and L?

I2: Would it be possible to add to Section 3 some intuition of why these are the appropriate functional forms for the IMED indices - they may seem somewhat arbitrary to the reader unfamiliar with Honda and Takemura (2015).

I3: Is the S in Algorithm 3 the same S as used to bound $\theta^*$? Should it be set as a ceiling or floor of $log(T)$ so that $[S]$'s make sense, and does line 15 of Alg 3 also apply for $s' \in [S]$?

I4: It would be useful to explain to the reader (or would be practitioner) how they should decide between the four algorithms presented when facing a practical problem.

I5: Lemma 1 should have some more extensive conditions? I believe in Abassi-Yadkori it is required that $V_t$ is positive definite?

S1: Formatting in Alg 1 could be clearer, maybe making use of cases in lines 10-12 and lines 6-9

S2: Lots of brackets throughout are not sized adaptively to their inputs.

S3: The wording 'Step 1/2/3' in Algorithm 3 suggests that the algorithm will sequentially progress through the steps without returning. Is this the case, and if not would these be more usefully referred to as 'Cases 1, 2, 3' instead?

S4: On page 6, you say that the linear extension of IMED was 'unknown'. This isn't quite the right wording? Presumably there is more than one way to extend it (there are three in the paper), so it's the case that an extension of IMED 'with theoretical guarantees' had not been identified? or that an extension hadn't been considered in the literature?

S5: On page 6, you mention 'the term $1/T_i(t)$ but eq(1) has a $log(T_i(t))$?

S6: I think an L^2 appears as a typo on p8

S7: There is a suppression of dependence on $\gamma$ in the proof sketch and around eq (15) in the appendix: this should be indicated to the reader more clearly I feel.

S8: shouldn't (16) lead to $O(\sqrt{dT}\log^{3/2}(T))$ also, once you account for $\beta_T$?

**Strengths And Weaknesses:**

The strengths of the paper are that it fills a genuine gap in the literature: the extension of a popular and effective family of algorithms for the multi-armed bandit problem to the linear case, and that the theoretical work is extensive and (to the best of my ability to judge in the time-frame) accurate. To elaborate, the IMED algorithms are, as the authors nicely summarise, successful approaches for the multi-armed bandit problem, particularly for reasons such as their capacity to be more easily incorporated into settings where counterfactual offline evaluation is necessitated. It is therefore a clear benefit to explore the extension of these methods to the linear bandit problem (and in future work, beyond). The theoretical work derives strong results for each of the algorithms, improving over what is achievable for linear Thompson Sampling, matching the best finite-time performance of optimistic algorithms, while maintaining an improved computational efficiency over these optimistic approaches. The layout of these results is mostly reader-friendly, with a detailed proof sketch for the first algorithm provided in the main text and explaining the key steps and the novelties therein, and the more detailed aspects relegated to appendices.

The most substantial weakness is in the experimental evaluation, specifically around the comparison to alternative approaches, and the choice of hyper parameters.
1. I found the justification for not including SupLinIMED and other SupLin algorithms not to be the strongest. Why not just include these to illustrate this point and show the reader how much worse they actually are?
2. You mention a similarity to Information Directed Sampling, and its asymptotic optimality - why not include it in your empirical comparison as a stronger benchmark?
3. It is unclear whether the multiplication by $\alpha$ breaks the theoretical guarantees of the previous section, and how the algorithms would perform if you had tested the version with the theoretical guarantees. It also seems curious to tune $\alpha$ over $T=100$ rounds when the algorithms are run for much longer in practice.
4. Why are very different horizons used across the various experiments?
5. A precise explanation of what is shown in the plots is not quite given. There is a remark in the text that states a mean and standard deviation are computed, so I infer that they show means plus and minus one standard deviation, however it is not clear whether the distributions of regret are symmetric as this would imply. Thompson Sampling can, for instance, often lead to skewed distributions.
6. I was a bit confused as to why repeated the suboptimal arms in the experiment of 6.1 makes any difference to the IMED algorithms, when there is a common $\theta^*$ will the algorithm not perform the same as if there was only one of these suboptimal arms.
7. Does the experimental setup allow $||\theta^*||$ and $||x_t||$ to exceed the assumed bounds also?
8. You speak to a statistical significant difference regardless of $d$ or $K$. This doesn't seem obvious in, for instance, Fig 2 (c)?
9. Building on point 7, it seems that the LinIMED algorithms enjoy some favourable selection of its parameters (in ways that may break (or alter) the associated theoretical guarantees) but it is not clear whether this occurred for the TS and UCB variants?

---

> ### Author Response · Authors · 2024-03-18
>
> $\textbf{Clarification on point 1: }$Thanks for the suggestion. For completeness, we have added SupLinIMED and SupLinUCB curves to Figures 1 and 2. It is evident from these figures that since SupLin-type algorithms explore too much initially , their regrets are not comparable to the more popular algorithms such as LinUCB.
>
> $\textbf{Clarification on point 2: }$Thanks for your valuable suggestions, we have added a qualitative comparison of our algorithms to the IDS in Section 3.3 in the revised paper. We also have also shown how IDS performs empirically in Figures 3 and 4 since to the best of our knowledge, IDS is designed for fixed and finite arm set setting.
>
> $\textbf{Clarification on point 3: }$Multiplication by $\alpha$ will result in a constant to the regret bounds. For the synthetic dataset, we do not use this tuning parameter for all the algorithms; we retain the use of $\beta_t(\gamma)$ as the confidence radius. For the MovieLens dataset, we ensure that we tune this $\alpha$ parameter for **all** the algorithms. So all the comparisons are fair. We tune $\alpha$ over the first $T_0=1000$ rounds as we treat these initial rounds as the "training dataset". The chosen $\alpha$ is then used for the remaining $T-T_0$ rounds where $T$ is the horizon. Details of the tuning of $\alpha$ are provided in Appendix G.
>
> $\textbf{Clarification on point 4: }$In the revised paper, we have used the same horizon for all the experiments for the same experiment.
>
> $\textbf{Clarification on point 5: }$Yes, the error bars, represented by the shaded area, in our figures refer to the mean plus/minus one standard deviation over the 100 independent trials.
>
> $\textbf{Clarification on point 6: }$Thanks for your careful observations. We agree that repetative arms do not provide extra information in linear bandit setting, so we have changed the instance for Figures 1 and 2 such that now the feature vector of each suboptimal arms now is now perturbed slightly by some indentical and independent uniform distributed noise over time. To be more specific, the feature vector of the suboptimal arms except the worst arm now is $[(1-\frac{1}{7+z_{t,i}})\frac{1}{\sqrt{d-1}}, (1-\frac{1}{7+z_{t,i}})\frac{1}{\sqrt{d-1}}, ... , (1-\frac{1}{7+z_{t,i}})\frac{1}{\sqrt{d-1}}, (1-\frac{1}{7+z_{t,i}})]^T\in \mathbb{R}^d$ where $z_{t,i}\sim \textrm{Uniform}[0,0.1]$ is  i.i.d. noise for each $t \in [T], i\in [K-2]$.
>
> $\textbf{Clarification on point 7: }$Yes. In the empirical study, $\theta^*$ and $\|x_t\|$ do not have to satisfy the bounds for the theoretical results; the assumptions are merely for proving the theoretical results. In addition, in our empirical study, we shrink the confidence interval $\sqrt{\beta_{t-1}(\gamma)}$ by some constant $\alpha\le 1$. Then we tune $\alpha$ on the grid search $[0.05, 0.1, 0.15, 0.2, ... , 0.95, 1.0]$ for all the algorithms to find the best performance of each algorithm. This hyperparameter tuning procedure (detailed in Appendix G) is usually performed in practical application of LinUCB and LinTS since the confidence bound proved in the theoretical guarantees in LinUCB and LinTS are conservative for the empirical study.
>
> $\textbf{Clarification on point 8: }$For the problem instance considered in Figure 2, the reviewer can observe (from Figure 2c) now there is a statistical significant gap between LinIMED-3 and LinUCB/LinTS for all $d$ or $K$.
>
> $\textbf{Clarification on point 9: }$For all the algorithms considered in the experiments, we tuned their confidence radii to ensure that the comparisons are fair. The theoretical guarantees, however, hold for certain choices of the confidence radii $\beta_t(\gamma)$; these may not lead to the best empirical performance, as it is well known. Hence, hyperparameter tuning is still required. We perform the hyperparameter tuning for all algorithms in systematically.
>
> $\textbf{Response to request change I1: }$ We can show the explicit dependence of the expected cumulative regrets on $S$, $L$, and $R$. Take LinIMED-2 as an example. Notice that in Lemmas 6, 7, 8 we have upper bounds of each case with respect to the term $\beta_T$, so if one substitutes $\beta_T=(R\sqrt{d\log(\frac{1+TL^2/\lambda}{\gamma})}+\sqrt{\lambda}S)^2$ and $\gamma=1/t^2$ into $R_t=O(\sqrt{dT\beta_T}\log (TL^2))$ then one can easily get a regret bound that depends on $S$, $L$ and $R$. In fact, one can also bound the gaps with a free parameter in Assumption 1. This is tantamount to changing the horizon of $\Delta_t$ in our peeling device and following the original proof. For simplicity, we bound the gap just by 1 because that is the convention in linear bandit setting; cf. Assumption 19.1 (b) in [1].
>
> [1] Lattimore, Tor, and Csaba Szepesvári. Bandit algorithms. Cambridge University Press, 2020.

---

> ### Author Response · Authors · 2024-03-18
>
> $\textbf{Response to request change I2: }$We have provided some intuition of the IMED term that is designed for the linear bandit setting in the last few sentences in Section 3.2. Briefly, this is because the KL terms reduce to the square of the empirical gaps. Besides, we have added the relation of our LinIMED to IDS in Section 3.3, the relationship to information also justifies our choice of the functional forms for the IMED indices.
>
> $\textbf{Response to request change I3: }$Thanks for your point, we have change $S$ in the SupLinIMED algorithm to $S'$ to make it different from the bound S in Assumption 1. We have apply a ceiling to $\log T$ to make it safe for use in $[S']$. Yes the line 15 of ALg 3 also applies for $s' \in [S']$.
>
> $\textbf{Response to request change I4: }$LinUCB, LinTS, and LinIMED-3 are recommended. To further decide the best one to use, the user can apply offline evaluation techniques like inverse propensity score in [1] to find the best solution when faced with a practical problem.
>
> [1]Miroslav Dudík, John Langford, and Lihong Li. 2011. Doubly robust policy evaluation and learning. arXiv preprint arXiv:1103.4601 (2011).
>
> $\textbf{Response to request change I5: }$Notice that in the algorithm we have additional $\lambda I$ to ensure that the design matrices $\{V_t\}$ are positive semidefinite.
>
>
> $\textbf{Response to request change S1: }$We have separated the cases with respect to the mode value $x\in\{1, 2, 3\}$ of LinIMED-x to make the formatting in Alg 1 much clearer.
>
> $\textbf{Response to request change S2: }$We have fixed this issue such as in the statement of the algorithms.
>
> $\textbf{Response to request change S3: }$Thanks for your suggestion. We have changed it to Cases 1/2/3.
>
> $\textbf{Response to request change S4: }$To the best of our knowledge, we are the first to extend IMED to linear bandit setting with a near-optimal minimax regret. We have modified the wording according to your suggestions on page 5 now:"However, an extension of IMED algorithm with minimax regret bound of
>  $\widetilde{O}(d\sqrt{T})$ has not been derived".
>
> $\textbf{Response to request change S5: }$In fact, that is not a typo actually. We regard $1/T_i(t)$ as the variance of arm $i$ at time $t$. Consequently, the second term in Equation (1), namely $\log(T_i(t))$, corresponds to the negative log of the variance of arm $i$ at time $t$. To explain this in more detail, let's take LinIMED-1 as an example. The variance of arm $i$ at time $t$ is precisely ${\beta_{t-1}(\gamma) \lVert x_{t,a} \rVert_{V_{t-1}^{-1}}^2}$, hence the second term of LinIMED-1 should be $-\log ({\beta_{t-1}(\gamma) \lVert x_{t,a} \rVert_{V_{t-1}^{-1}}^2})$, which is exactly what it is.
>
> $\textbf{Response to request change S6: }$We have corrected this typo in the sentences between equations (2) and (3), the modified text now is ''since the largest eigenvalue of the matrix the matrix $V_{t-1}$ is upper bounded by $\lambda+TL^2$''.
>
> $\textbf{Response to request change S7: }$We have added some texts around and in Eqn. (15) such as the "peeling device" on $\Delta_t$ such that $2^{-l} < \Delta_t \le 2^{-l+1}$ for $l=1,2,\ldots,\lceil Q\rceil$ where $Q=-\log_2\Gamma$ " to make it more reader-friendly.
>
> $\textbf{Response to request change S8: }$Thanks for pointing out this typo. It is indeed a typo and should lead to the regret bound $O(d\sqrt{T}\log^{\frac{3}{2}}T)$. We have fixed this in the revised version of the manuscript.

---

> > ### Comment · Reviewer_CTEL · 2024-03-25
> > **Reply to Author Response**
> >
> > Thank you authors for your systematic response. In short, I will be recommending the paper for acceptance, as I think there is now a good coverage of the important issues and the experimental section has improved.

---

> > > ### Author Response · Authors · 2024-03-25
> > > **Reply to Reviewer CTEL**
> > >
> > > Dear Reviewer CTEL,
> > >
> > > Thanks for your recommendation of acceptance of our paper.
> > >
> > > Authors

---

### Review · Reviewer_hHYa · 2024-02-16

**Summary Of Contributions:**

This work studies the question of regret minimization in linear bandits. They propose an algorithm inspired by the multi-armed bandit IMEB algorithm, and prove that it achieves nearly-minimax optimal regret. Experimental results demonstrate that it yields small improvements over LinUCB and LinTS.

**Audience:**

Yes

**Broader Impact Concerns:**

None.

**Claims And Evidence:**

No

**Requested Changes:**

- The claim that the algorithm is computationally efficient should be removed (typically a linear bandit algorithm is considered computationally efficient if it can be reduced to efficient computation oracles, such as a linear maximization oracle).
- The presentation could be cleaned up somewhat. The paper presents four algorithms total, and it is not clear what the advantage of one over the other are. For example, LinIMED-1 achieves the same regret bound as LinIMED-2, while requiring stronger assumptions. This makes the statement of the algorithm somewhat confusing. My suggestion would be to remove LinIMED-1 and LinIMED-2, or put them in the appendix, and only keep LinIMED-3 and SupLinIMED (and their corresponding theorems) in the main text.
- The confidence intervals in Figures 1, 2, and 4 overlap significantly so it is not entirely clear which algorithm is the best or worst. The number of trials should be increased so there is greater separation between the approaches.
- In addition to including more discussion comparing LinIMED to IDS, it would also be helpful to include empirical results for IDS.
- In Figure 3, it would be helpful to also compare against an algorithm known to be asymptotically optimal (for example the asymptotically optimal variant of IDS mentioned in section 6.2).
- A full proof is missing from the paper—only a proof sketch for Theorem 1 is given. Proofs for all results must be included (at least in the appendix).

Minor typos:
- The definition of regret given in the intro is incorrect—rather the max should be over the expected mean for each arm $a$ rather than, $Y_{t,a}$, the random reward received.
- Some of the indices in Algorithm 1 are incorrect. For example, on line 9, the argmax is over $j$, but the objective of the argmax uses $a$.

**Strengths And Weaknesses:**

Strengths:
- The algorithm is novel in the linear bandit setting (to the best of my knowledge) and achieves nearly minimax-optimal regret.


Weaknesses:
- The authors claim that their approach is computationally efficient for large finite arm sets. I do not believe this is the case, however, as their algorithm involves computing an index for each arm, so the computational complexity will scale with $K$, the number of arms. This claim should be revised.
- Algorithm 1 is extremely similar to information-directed sampling (see in particular the variants presented in [1] and [2]). More discussion of this should be given.
- Many algorithms which achieve minimax optimal regret for linear bandits are already known. While technically the algorithm here is novel, it does not achieve a result any better than existing works, and, as noted, its techniques are very similar to existing works.


[1] Kirschner, Johannes, and Andreas Krause. "Information directed sampling and bandits with heteroscedastic noise." Conference On Learning Theory. PMLR, 2018.

[2] Kirschner, Johannes, Tor Lattimore, and Andreas Krause. "Information directed sampling for linear partial monitoring." Conference on Learning Theory. PMLR, 2020.

---

> ### Author Response · Authors · 2024-03-18
>
> $\textbf{Response to Weakness 1: }$
> Thanks for pointing out this issue. The computational complexity indeed scales with $K$ but this is the same as the LinUCB algorithm using OFUL's confidence region. We have removed the claim in Table 1 that LinIMED is computationally efficient for large finite arm sets.
>
> $\textbf{Response to Weakness 2: }$
> Our LinIMED algorithms are similar to IDS for the first term, the difference is that we have additional second term in our index $-\log( \beta_{t-1}(\gamma)\lVert x\_{t,a}\rVert^2\_{V^{-1}\_{t-1}})$  to encourage exploration. We have added a paragraph comparing LinIMED and to the IDS papers in the newly created Section 3.3. Please check that paragraph for more details.
>
> [1] Kirschner, Johannes, and Andreas Krause. "Information directed sampling and bandits with heteroscedastic noise." Conference On Learning Theory. PMLR, 2018.
>
> [2] Kirschner, Johannes, Tor Lattimore, and Andreas Krause. "Information directed sampling for linear partial monitoring." Conference on Learning Theory. PMLR, 2020.
>
> $\textbf{Response to Weakness 3: }$ Regret minimization for linear bandit is indeed well-studied and many famous algorithms have been proposed. Our techniques are similar to some existing works because the designs of our algorithms are simple and there are ample proof techniques in linear bandit area. What we have done in this paper is to show that the linear bandit extension based on the principle of minimum empirical divergence also matches ubiquitoius linear bandit algorithms like LinUCB and LinTS. We have also shown that our algorithms performs better in the "End of Optimism" instance and some real-world dataset like the MovieLens dataset.
>
> $\textbf{Response to Requested Change 1: }$ We have removed that claim in Table 1 in our paper.
>
> $\textbf{Response to Requested Change 2: }$ We prefer to keep the discussion of LinIMED-1 and LinIMED-2 in the main paper since they reveal the story of our designing the extension of IMED for linear bandits and we believe understanding the intuition behind their simpler design is beneficial for future work. Although there is an additional assumption for the regret bound of LinIMED-1, it is the version that most similar to the structure of IMED algorithm. However, since linear bandit setting is more structured than the $K$-armed bandit setting, we came up with LinIMED-2 to further encourage exploitation. Additionally, in LinIMED-2 we can relax the assumption that is made to guaranteed the regret bound of LinIMED-1. Finally, to remove the extraneous $\sqrt{\log T}$ term in the regret bound of LinIMED-2, we further furnish the upper bound of the index with the greedy action and propose LinIMED-3 whose regret bound matches state-of-art algorithms like LinUCB.
>
> $\textbf{Response to Requested Change 3: }$ The large error bars are due in part to the fact that LinTS (and the newly included IDS) are randomized algorithms. We have increased the number of trials in our experiments (up to the budget of our computational resources) and all the confidence intervals have been reduced. See the newly revised figures.
>
> $\textbf{Response to Requested Change 4: }$ Thanks for your valuable suggestions. We have added the comparison of LinIMED to IDS in Section 3.3 in our paper. The empirical results for IDS are added to Figures 3 and 4. To the best of our knowledge, IDS is designed for the fixed finite arm set setting, so Figures 1 and 2 do not include the curves representing the performances of IDS since the settings there are for varying arm sets.
>
> $\textbf{Response to Requested Change 5: }$  We have added the empirical result of the asymptotically optimal algorithm IDS mentioned in Figure 3 of Section 6.2. One observes that our LinIMED-3 is comparable to IDS for these "End of Optimism" instances. This is pleasing for LinIMED as IDS is asymptotically optimal, but our LinIMED algorithms are designed to be worst-case optimal, yet work well on these "End of Optimism" instances.
>
> $\textbf{Response to Requested Change 6: }$ We have presented the full proofs in the appendix, please download the supplemental zip file to see the appendix of our paper. We will also make available all the code to reproduce our plots in the final (accepted) version of this paper.
>
> $\textbf{Response to Minor typos: }$ Thanks for your notice, we have fixed these typos.

---

### Review · Reviewer_DYKa · 2024-02-26

**Summary Of Contributions:**

In this study, they proposed the first Indexed Minimum Empirical Divergence (IMED)-based algorithm for linear bandits and provide \tilde{O}(d \sqrt{T})-regret bound where d is the dimension. The algorithms can deal with time-varying action space A_t at each round t. They also consider SupLinIMED, a variant of SupLinICB (Chu et al. 2011) with regret bound depending on the number of arms K. A series of experiments are provided to demonstrate that proposed methods work better than LinUCB or LinTS-based methods.

**Audience:**

Yes

**Claims And Evidence:**

Yes

**Requested Changes:**

Could you try to add a discussion on the comparison with the related work mentioned above and other results in the context of adversarial linear bandits?

**Strengths And Weaknesses:**

Strength
1. The first linear extension of the IMED algorithm.
2. Experiments confirmed that LinIMED significantly improves on LinUCB and Linear Thompson Sampling in the End of Optimism instance. Moreover, experiments with real data using MovieLens were also conducted.
3. The paper is easy to follow and well-structured and explained in a short length.

Weakness

1. The LinIMED itself seems to be a fairly simple method since it uses the similar confidence intervals of [Abbasi-Yadkori et al. 2011] and essentially the UCB is instead changed to IMED, which are not many novel techniques. The theoretical analysis is also just an application of ideas already used in existing studies.

2. Missing related work:
Indexed Minimum Empirical Divergence for Unimodal Bandits (NeurIPS 2021)
Hassan SABER, Pierre Ménard, Odalric-Ambrym Maillard
https://proceedings.neurips.cc/paper/2021/hash/3c88c1db16b9523b4dcdcd572aa1e16a-Abstract.html

3. No distribution-dependent regret bound is provided. Also, there exists a good summary of adversarial linear bandits by [Liu+NeurIPS2023]:
 Bypassing the Simulator: Near-Optimal Adversarial Linear Contextual Bandits
Haolin Liu · Chen-Yu Wei · Julian Zimmert

Would it be great if the authors could add a comparison with this literature, though the submitted paper considers stochastic loss and adversarial action sets? ([Liu+2023] focuses on stochastic action sets, if I am not missing)

Question: Asymptotically optimal is not achieved, but where do you see the difficulty in achieving it?

---

> ### Author Response · Authors · 2024-03-18
>
> $\textbf{Response to Weakness 1}$: Thanks for your review, we agree that the theoretical analysis is an application and combination of existing techniques. This is due to the fact that linear bandit is an widely studied area. However, the design of our algorithm is simple and does not include some term that as complicated as the index as in asymptotically optimal information-directed sampling [1]. Our objective is to extend the IMED algorithm for K-armed bandits so that it is amenable to linear bandits. Our extension adheres to the IMED principle, retains the minimax optimality of more common algorithms, and works well empirically.
>
> $\textbf{Response to Weakness 2}$: Thanks for your suggestions, we have added the comparison of our algorithms to the IMED for unimodal bandits in Appendix.B in our revised paper.
>
> $\textbf{Response to Weakness 3}$:  We have added the comparison of this novel adversarial linear bandit algorithm with our LinIMED algorith in Section Appendix.B in our paper. For the lack of distribution-dependent regret bound, basically it is hard to show the asymptotically optimality in linear bandit for deterministic algorithm since the definiton of asymptotical lower bound in linear bandit is on solving an optimization problem for the distribution of the arm pulls (allocation). Please see the detailed reply to your Question 1 below.
>
>
> [1] Kirschner, Johannes, et al. "Asymptotically optimal information-directed sampling." Conference on Learning Theory. PMLR, 2021.
>
> $\textbf{Response to Question 1}$: Achieving asymptotic optimality in the linear bandit setting is different from in $K$ armed bandit setting. Recall the definition of lower bound of linear bandit in the IDS paper:
>
> For an allocation $\alpha \in \mathbb{R}^{\mathcal{X_{\ge 0}}}$ over actions we define the associated coriance matrix $V(\alpha)=\sum_{x\in \mathcal{X}}\alpha(x)xx^T$. Let $c^*$ be the solution to the following convex program,
> $$c^*:=\inf_{\alpha\in\mathbb{R}^{\mathcal{X_{\ge0}}}}\sum_{x\in \mathcal{X}}\alpha(x)\langle x^*-x,\theta^* \rangle  \quad\text{s.t.}\quad \min_{v\in \mathcal{C}^*} \frac{1}{2}\lvert\lvert v-\theta^* \rvert\rvert^2_{V(\alpha)}\ge 1$$
>
>  the lower bound of linear bandit with fixed arm set related to the optimization of allocation of arms (distribution of choosing the arms), which is different from that in $K$-armed bandit setting we have exact closed-form instance dependent lower bound for certaion reward distributions.
> The difficulty is on showing an analogue of the "asymptotic information ratio Lemma" as in the IDS paper (Lemma 11 therein). Since the asymptotically optimal IDS algorithm is based on minimizing the IDS ratio with a distribution over arm pulls:
> $$\pi_t^{\mathrm{IDS}}:=\arg\min_{\pi\in \mathcal{D}(\mathcal{A})} \frac{\hat{\Delta}_t^2(\pi)}{g_t(\pi)},$$
>  the IDS ratio naturally is directly related to the definition of the lower bound of linear bandit (the allocation of arms $\alpha(x)$). However, our algorithm is deterministic, which means at each time step our algorithm chooses the arm with the minimum index rather than sampling from an optimized distribution:
> $$\pi_t^{\mathrm{LinIMED1}}:=\arg\min\_{a\in \mathcal{A}_t} \frac{\hat{\Delta}\_{t,a}^2}{\beta_t(\delta) \lvert\lvert x\_{t,a}\rvert\rvert^2\_{V^{-1}\_{t-1}}}+\log (\frac{1}{\beta_t(\delta) \lvert\lvert x\_{t,a}\rvert\rvert^2\_{V^{-1}\_{t-1}}}).$$
> Because of this reason,   it is hard to show an "asymptotic information ratio lemma" since our algorithm is not based on optimizing for a distribution over the arms. One potential future direction is to find a randomized version of the LinIMED algorithm such that it also minimizes some index with respect to the allocation of arms $\alpha(x)$ and see whether it achieves asyptotic optimality.
>
> $\textbf{Response to Requested Change}$: Thanks for your suggestions, as mentioned before, we have added the comparison with the related work mentioned in Sections 3.3 and Appendix.B in the revised paper.

---

### Decision · Action_Editor_YEtu · 2024-03-31

**Recommendation:** Accept as is

**Comment:**

This paper introduces the Linear Indexed Minimum Empirical Divergence (LinIMED) algorithms, a novel extension of the Indexed Minimum Empirical Divergence (IMED) approach, to the linear bandit setting. The proposed algorithms, including LinIMED and its variant SupLinIMED, aim to fill a gap in the literature by providing a computational and theoretical advancement over existing methods. The authors offer a comprehensive theoretical analysis, demonstrating that LinIMED algorithms achieve a $\widetilde{O} (d\sqrt{T})$ regret bound, where $d$ is the context dimension and $T$ is the time horizon. Empirical studies further validate the superior performance of LinIMED over popular linear bandit algorithms such as LinUCB and Linear Thompson Sampling in various settings.

The paper's major strength lies in its pioneering theoretical contribution and practical effectiveness, filling a notable gap in the literature on linear bandits. The introduction of LinIMED algorithms and the demonstration of their superior performance through rigorous theoretical analysis and extensive empirical validation highlight the paper's significant impact.

**Audience:**

Yes

**Claims And Evidence:**

Yes